# Directional Pruning of Deep Neural Networks

**Shih–Kang Chao**[*]
Department of Statistics
University of Missouri
Columbia, MO 65211
chaosh@missouri.edu

**Zhanyu Wang**
Department of Statistics
Purdue University
West Lafayette, IN 47907
wang4094@purdue.edu

**Yue Xing**
Department of Statistics
Purdue University
West Lafayette, IN 47907
xing49@purdue.edu

**Guang Cheng**
Department of Statistics
Purdue University
West Lafayette, IN 47907
chengg@purdue.edu

## Abstract

In the light of the fact that the stochastic gradient descent (SGD) often finds a flat minimum valley in the training loss, we propose a novel directional pruning method which searches for a sparse minimizer in or close to that flat region. The proposed pruning method does not require retraining or the expert knowledge on the sparsity level. To overcome the computational formidability of estimating the flat directions, we propose to use a carefully tuned $\ell_1$ proximal gradient algorithm which can provably achieve the directional pruning with a small learning rate after sufficient training. The empirical results demonstrate the promising results of our solution in highly sparse regime (92% sparsity) among many existing pruning methods on the ResNet50 with the ImageNet, while using only a slightly higher wall time and memory footprint than the SGD. Using the VGG16 and the wide ResNet 28x10 on the CIFAR-10 and CIFAR-100, we demonstrate that our solution reaches the same minima valley as the SGD, and the minima found by our solution and the SGD do not deviate in directions that impact the training loss. The code that reproduces the results of this paper is available at `https://github.com/donlan2710/gRDA-Optimizer/tree/master/directional_pruning`.

## 1 Introduction

Deep neural networks (DNNs), after properly trained, provide the state-of-the-art performance in various domains. Overparameterization is a common practice in modern deep learning, which facilitates better expressive power and faster convergence. On the other hand, overparameterization makes DNN exceedingly large, especially for large-scale tasks. For example, the ImageNet [10, 52] may need billions of parameters [4] to become sufficiently overparameterized. As the number of parameters in DNN is growing fast, the cost to deploy and process large DNNs can be prohibitive on devices with low memory/processing resources or with strict latency requirements, such as mobile phones, augmented reality devices and autonomous cars. Many achievements have been made in shrinking the DNN while maintaining accuracy, and the MIT Technological Review lists the "tiny AI" as one of the breakthroughs in 2020 [1].

Among many methods for shrinking DNN, sparse DNN has attracted much attention. Here, sparsity refers to the situation that most model parameters are zero in a DNN. Sparse DNN not only requires

---

[*]Corresponding author.

less memory and storage capacity, but also reduces inference time [9]. One of the popular ways to get sparse DNNs is magnitude pruning [27, 26, 43, 62, 40, 17, 18, 19]. Magnitude pruning first learns the model parameters with an optimizer, e.g. stochastic gradient descent (SGD), and then prunes based on the learned magnitude of parameters with an a priori threshold. However, determining a threshold requires some expert knowledge and trial-and-error, as a principle for setting the threshold is not available. In addition, naïvely masking parameters usually worsens the training loss and testing accuracy. Hence, retraining is needed for the pruned network to regain a similar performance as the dense network [27]. Unfortunately, retraining as an additional step requires some care [17] and additional computation.

## 1.1 Directional pruning

In this paper, we try to answer when a coefficient can be pruned without paying the price of increasing the training loss, and how we can prune based on this. These answers rely on the local geometry of the DNN loss function $\ell(\mathbf{w})$, where $\mathbf{w}$ denotes the parameters.

Suppose that $\mathbf{w}^{SGD} \in \mathbb{R}^d$, the parameter trained by the SGD, has reached a valley of minima. Hence, $\nabla \ell(\mathbf{w}^{SGD}) \approx 0$. The Hessian $\nabla^2 \ell(\mathbf{w}^{SGD})$ has multiple nearly zero eigenvalues [53, 54, 21, 48], and the directions associated with these eigenvalues are the flat directions on the loss landscape. Perturbation in these directions causes little change in the training loss by the second order Taylor expansion of $\ell(\mathbf{w})$ around $\mathbf{w}^{SGD}$. We denote the subspace generated by these directions as $\mathcal{P}_0$.

Following [37, 29], pruning $\mathbf{w}^{SGD}$ can be viewed as a perturbation of $\mathbf{w}^{SGD}$:

$$\mathbf{w}^{SGD} - A \cdot \text{sign}(\mathbf{w}^{SGD}). \tag{1}$$

Here, $\text{sign}(\mathbf{w}^{SGD}) \in \{-1, 1\}^d$ is the sign vector of $\mathbf{w}^{SGD}$ and $A$ is a diagonal matrix with $0 \leq A_{jj} \leq |w_j^{SGD}|$ for $j = 1, \ldots, d$. The $j$th coefficient is pruned if $A_{jj} = |w_j^{SGD}|$. For example, in a 2D illustration in the left panel of Figure 1, (1) is a vector starting from the origin to a point in the orange rectangle.

Retraining is needed if $A \cdot \text{sign}(\mathbf{w}^{SGD}) \notin \mathcal{P}_0$. Some empirical studies even suggest $\mathcal{P}_0$ is nearly orthogonal to the $\mathbf{w}^{SGD}$ [25, 21], so generally $A \cdot \text{sign}(\mathbf{w}^{SGD}) \notin \mathcal{P}_0$. Therefore, we instead consider $\mathbf{w}^{SGD} - \lambda \cdot \Theta$ where the perturbation direction $\Theta \in \mathcal{P}_0$ and $\lambda > 0$. We maximize the number of $j$ such that $\text{sign}(\Theta_j) = \text{sign}(w_j^{SGD})$ for $j = 1, \ldots, d$, in order to decay as many coefficients in $\mathbf{w}^{SGD}$ as possible. Specifically, we select $\Theta$ as

$$\Theta = \arg\min_{\mathbf{u} \in \mathcal{P}_0} \left\| \mathbf{u} - \text{sign}(\mathbf{w}^{SGD}) \right\|_2^2,$$

i.e. $\Theta = \Pi_0\{\text{sign}(\mathbf{w}^{SGD})\}$, where $\Pi_0$ denotes the projection on the subspace $\mathcal{P}_0$. The vector $\Theta$ does not always decrease the magnitude of $\mathbf{w}^{SGD}$, and it does whenever $\text{sign}(w_j^{SGD}) \cdot \Theta_j > 0$, or

$$s_j := \text{sign}(w_j^{SGD}) \cdot \left( \Pi_0\{\text{sign}(\mathbf{w}^{SGD})\} \right)_j > 0. \tag{2}$$

Decreasing the magnitude of the coefficients with $s_j > 0$ in $\mathbf{w}^{SGD}$ would cause little changes in the training loss, as long as we simultaneously increase the magnitude of coefficients $j' \neq j$ with $s_{j'} < 0$ proportional to $|s_{j'}|$. As illustrated in the left panel of Figure 1, the adverse effect due to decreasing the magnitude of $w_2$ ($s_2 > 0$) can be compensated by increasing the magnitude of $w_1$, so that the net change is the red vector in $\mathcal{P}_0$. Note that this argument has a similar spirit as the "optimal brain surgeon"[29], and it is the key to remove the need of retraining. The $s_j$ can thus be understood as a score to indicate whether pruning the $j$th coefficient causes a (ir)redeemable training loss change. We propose the novel "directional pruning" using the score $s_j$ in (2).

**Definition 1.1** (Directional pruning based on SGD). Suppose $\ell(\mathbf{w})$ is the training loss, and $\nabla \ell(\mathbf{w}^{SGD}) = 0$ where $\mathbf{w}^{SGD}$ is the minimizer found by SGD. Suppose none of the coefficients in $\mathbf{w}^{SGD}$ is zero. With $\lambda > 0$ and $s_j$ defined in (2), the directional pruning solves

$$\arg\min_{\mathbf{w} \in \mathbb{R}^d} \frac{1}{2}\|\mathbf{w}^{SGD} - \mathbf{w}\|_2^2 + \lambda \sum_{j=1}^{d} s_j |w_j|. \tag{3}$$

In (3), the coefficients with $s_j > 0$ are pruned with sufficiently large $\lambda$ by the absolute value penalization, but the magnitude of $w_{j'}$ with $s_{j'} \leq 0$ is un-penalized, and are even encouraged to

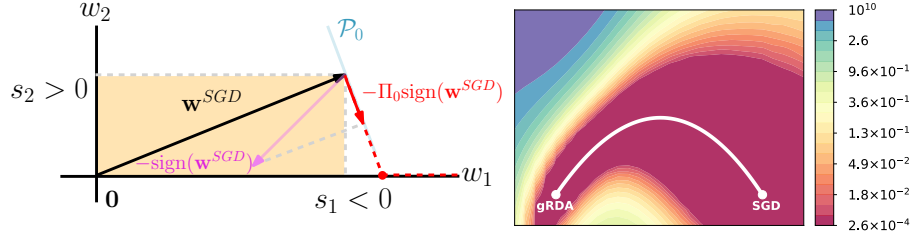

Figure 1: **Left:** a 2D graphical illustration of the directional pruning. The orange region contains all possible locations of the vector $\mathbf{w}^{SGD} - A \cdot \text{sign}(\mathbf{w}^{SGD})$. The directional pruning with different $\lambda$ takes solutions on the red dashed line. **Right:** training loss contour of the wide ResNet28×10 (WRN28x10 [60]) on the CIFAR-100 around the minimal loss path (the white curve) between minimizers found by the SGD and (gRDA) [8] (the algorithm we propose to use) using [20]. While no coefficient of the SGD minimizer is zero, our solution has only 9.7% active parameters. Testing accuracy is 76.6% for the SGD and 76.81% for our solution.

increase. For a 2D illustration, the solution path for different $\lambda > 0$ is the dashed red curve in the left panel of Figure 1. If $\lambda$ is too large, the coefficients $j$ with $s_j < 0$ may overshoot, illustrated as the flat part on the dashed red line extended to the right of the red point.

**Remark 1.2** (Solution of (3)). The objective function in (3) is separable for each coefficient. The part with $s_j > 0$ is solved by the $\ell_1$ proximal operator. The part with $s_j < 0$ is non-convex, but it still has the unique global minimizer if $w_j^{SGD} \neq 0$. The solution of (3) is

$$\widehat{w}_j = \text{sign}(w_j^{SGD})\big[|w_j^{SGD}| - \lambda s_j\big]_+,$$

where $[a]_+ = \max\{0, a\}$. See Proposition A.1 in the appendix for a proof.

Implementing the directional pruning is very challenging due to high dimensionality. Specifically, the matrix $\nabla^2 \ell$ of modern deep neural network is often very large so that estimating $\mathcal{P}_0$ is computationally formidable. Perhaps surprisingly, we will show that there is a very simple algorithm (gRDA) presented in Section 2, that can asymptotically solve (3) without explicitly estimating the Hessian. The right panel of Figure 1 shows that if $\lambda$ is selected appropriately, our method achieves a similar training loss as the dense network with $\mathbf{w}^{SGD}$, while being highly sparse with a test accuracy comparable to the SGD. More detailed empirical analysis is in Section 4.2.

**Remark 1.3** (Major differences to the "optimal brain surgeon"). It is worth noting that (3) is different from the optimization problem in [29, 28]. While an analytic map between directional pruning and optimal brain surgeon is interesting for future study, the two are generally nonequivalent. Particularly, directional pruning perturbs from $\mathbf{w}^{SGD}$ continuously in $\lambda$ like a restricted $\ell_1$ weight decay on $\mathcal{P}_0$ (Remark 1.2), while optimal brain surgeon yields a discontinuous perturbation like a hard thresholding (see p.165 of [29]). The main advantage of directional pruning is that it can be computed with the gRDA algorithm presented in Section 2, which does not require to estimate the Hessian or its inverse.

## 1.2 Contributions

Our major contribution is to propose the novel directional pruning method (Definition 1.1), and further prove that the algorithm (gRDA) [8] achieves the effect of the directional pruning asymptotically. The (gRDA) has been applied for sparse statistical inference problems with a convex loss and principal component analysis [8]. The connection between the directional pruning and (gRDA) is theoretically proved by leveraging the continuous time approximation developed in [8] under proper assumptions on the gradient flow and the Hessian matrix. It is worth noting that this algorithm does not require to explicitly estimate $\mathcal{P}_0$, and it can be implemented like an optimizer in a typical deep learning framework, e.g. Tensorflow or PyTorch.

Empirically, we demonstrate that (gRDA) successfully prunes ResNet50 on ImageNet, and achieves 73% testing accuracy with only 8% active parameters. Upon benchmarking with other popular algorithms, (gRDA) yields a high accuracy and sparsity tradeoff among many contemporary methods. We also successfully prune deep networks on CIFAR-10/100, and the results are in the appendix. Using VGG16 on CIFAR-10 and WRN28x10 on CIFAR-100, we show that (gRDA) reaches the

same valley of minima as the SGD, empirically verifying the directional pruning. Using VGG16 and WRN28x10 on CIFAR-10, we show the proportion of the difference between (gRDA) and the SGD in the leading eigenspace of the Hessian is low, as another evidence for (gRDA) performing the directional pruning.

## 2 The gRDA algorithm

Consider training data $Z_i = \{(X_i, Y_i)\}_{i=1}^N$, where $X_i$ is the input variable, e.g. images, and $Y_i$ is the response variable, e.g. a vector of real numbers, or labels $Y_n \in \{0, 1\}^{n_l}$, where $n_l \in \mathbb{N}$. Suppose $h(x; \mathbf{w}) \in \mathbb{R}^{n_l}$ is the output of an $L$-layer feedforward overparameterized DNN, with parameters $\mathbf{w} \in \mathbb{R}^d$. Let $\mathcal{L}(h; y) : \mathbb{R}^{n_l \times n_l} \to \mathbb{R}_+$ be a loss function, e.g. the $\ell_2$ loss $\mathcal{L}(h; y) = \|h - y\|_2^2$ or the cross-entropy loss. Let $f(\mathbf{w}; Z) := \mathcal{L}(h(X; \mathbf{w}), Y)$, and $\nabla f(\mathbf{w}; Z)$ be the gradient of $f(\mathbf{w}; Z)$, the loss function $\ell(\mathbf{w})$ and its gradient are defined by

$$\ell(\mathbf{w}) := \mathbb{E}_{\mathcal{Z}}[f(\mathbf{w}; Z)], \quad G(\mathbf{w}) = \nabla \ell(\mathbf{w}) = \mathbb{E}_{\mathcal{Z}}[\nabla f(\mathbf{w}; Z)], \tag{4}$$

where $\mathbb{E}_{\mathcal{Z}}[f(\mathbf{w}; Z)] = N^{-1} \sum_{i=1}^N f(\mathbf{w}; Z_i)$.

We adopt the generalized regularized dual averaging (gRDA) algorithms originally proposed in [8]. This algorithm has been successfully applied to the ad click-through rate prediction [39]. Specifically, let $\{\hat{i}_k\}_{k=1}^\infty$ be i.i.d. uniform random variables on $\{1, \dots, N\}$ independent from the training data,

$$w_{n+1,j} = \mathcal{S}_{g(n,\gamma)}\left(w_{0,j} - \gamma \sum_{k=0}^n \nabla f_j(\mathbf{w}_k; Z_{\hat{i}_{k+1}})\right), \text{ for } j = 1, \dots, d, \tag{gRDA}$$

where $\mathcal{S}_g : v \mapsto \text{sign}(v)(|v| - g)_+$ is the soft-thresholding operator, $\mathbf{w}_0$ is an initializer chosen at random from a distribution; $\gamma$ is the learning rate; $g(n, \gamma) > 0$ is the tuning function, detailed in (5). We can extend (gRDA) to minibatch gradients, by replacing $\nabla f_j(\mathbf{w}_k; Z_{\hat{i}_{k+1}})$ with an average $|S_{k+1}|^{-1} \sum_{i \in S_{k+1}} \nabla f(\mathbf{w}_k; Z_i)$, where $S_{k+1} \subset \{1, \dots, N\}$ is sampled uniformly. We will focus on (gRDA), i.e. $|S_k| = 1$ for all $k$, but our theory can be generalized to any fixed minibatch size.

The tuning function $g(n, \gamma)$ controls the growth rate of penalization. Motivated by [8],

$$g(n, \gamma) = c\gamma^{1/2}(n\gamma)^\mu, \tag{5}$$

where $c, \mu > 0$ are the two hyperparameters positively related to the strength of penalization. The $(n\gamma)^\mu$ is used to match the growing magnitude of SGD. The $\gamma^{1/2}$ is an important scaling factor; without it, (gRDA) with $\mu = 1$ reduces to the regularized dual averaging (RDA) algorithm [59] that minimizes $\ell(\mathbf{w}) + \lambda\|\mathbf{w}\|_1$ rather than the directional pruning problem in (3). Note that if $c = 0$, then (gRDA) recovers the stochastic gradient descent:

$$\mathbf{w}_{n+1}^{SGD} = \mathbf{w}_n^{SGD} - \gamma \nabla f(\mathbf{w}_n^{SGD}; Z_{\hat{i}_{n+1}}). \tag{SGD}$$

In this paper, we only consider the constant learning rate. In practice, a "constant-and-drop" learning rate is often adopted. See Section C.1 and C.2 in the appendix for the algorithms in pseudocode.

**Remark 2.1** (Selection of $\mu$ and $c$ in practice). Our empirical results and theory in later sections suggest $\mu \in \{0.501, 0.51, 0.55\}$ generally performs well regardless of the task and network used. For a given $\mu$, we recommend to search for the greatest $c$ (starting with e.g. $10^{-4}$) such that gRDA yields a comparable test acc. as SGD using $1 - 5$ epochs.

## 3 Theoretical analysis

To show (gRDA) asymptotically achieves the directional pruning in Definition 1.1, we leverage some tools from the continuous time analysis. Define the gradient flow $\mathbf{w}(t)$ to be the solution of the ordinary differential equation

$$\dot{\mathbf{w}} = -G(\mathbf{w}), \ \mathbf{w}(0) = \mathbf{w}_0, \tag{GF}$$

where $\mathbf{w}_0$ is a random initializer, and $G$ is defined in (4). The $\mathbf{w}(t)$ can provably find a good global minimizer under various conditions [3, 2, 13, 38, 46, 12]. Throughout this paper, we assume the solution of (GF) is unique.

Let $H(\cdot) := \mathbb{E}_{\mathcal{Z}}[\nabla^2 f(\cdot; Z)]$ be the Hessian matrix. Let $\Phi(t, s) \in \mathbb{R}^{d \times d}$ be the solution (termed the principal matrix solution, see Chapter 3.4 of [57]) of the matrix ODE system ($s$ is the initial time):

$$\frac{d\Phi(t, s)}{dt} = -H(\mathbf{w}(t))\Phi(t, s), \quad \Phi(s, s) = I_d. \tag{6}$$

Let $\mathbf{w}_\gamma(t) := \mathbf{w}_{\lfloor t/\gamma \rfloor}$ and $\mathbf{w}^{SGD}(t)$ be the piecewise constant interpolated process of (gRDA) and (SGD), respectively, with the same learning rate, where $\lfloor a \rfloor$ takes the greatest integer that is less than or equal to $a$. We will make the following assumptions:

**(A1)** $G(\mathbf{w}) : \mathbb{R}^d \to \mathbb{R}^d$ is continuous on $\mathbb{R}^d$.

Define

$$\Sigma(\mathbf{w}) := \mathbb{E}_{\mathcal{Z}}\big[\big(\nabla f(\mathbf{w}; Z) - G(\mathbf{w})\big)\big(\nabla f(\mathbf{w}; Z) - G(\mathbf{w})\big)^\top\big]. \tag{7}$$

**(A2)** $\Sigma : \mathbb{R}^d \to \mathbb{R}^{d \times d}$ is continuous. $\mathbb{E}_{\mathcal{Z}}\big[\sup_{\|\mathbf{w}\| \leq K} \big\|\nabla f(\mathbf{w}, Z)\big\|_2^2\big] < \infty$ for any $K > 0$ a.s.

**(A3)** $H : \mathbb{R}^d \to \mathbb{R}^{d \times d}$ is continuous, and there exists a non-negative definite matrix $\bar{H}$ such that $\int_0^\infty \|H(\mathbf{w}(s)) - \bar{H}\| ds < \infty$ where $\|\cdot\|$ is the spectral norm, and the eigenspace of $\bar{H}$ associated with the zero eigenvalues matches $\mathcal{P}_0$.

**(A4)** $\int_0^t s^{\mu-1}\Phi(t, s)\operatorname{sign}(\mathbf{w}(s))ds = o(t^\mu)$.

**(A5)** There exists $\bar{T} > 0$ such that for all $t > \bar{T}$: (i) $\operatorname{sign}\{\mathbf{w}(t)\} = \operatorname{sign}\{\mathbf{w}(\bar{T})\}$; (ii) $\operatorname{sign}\{w_j(t)\} = \operatorname{sign}\{w_j^{SGD}(t)\}$ for all $j$.

The key theoretical result of this paper shows that (gRDA) performs the directional pruning (Definition 1.1) for a sufficiently large $t$.

**Theorem 3.1.** Under assumptions (A1)-(A5), and assume $\mu \in (0.5, 1)$ and $c > 0$ in (5). Then, as $\gamma \to 0$, (gRDA) asymptotically performs directional pruning based on $\mathbf{w}^{SGD}(t)$; particularly,

$$\mathbf{w}_\gamma(t) \overset{d}{\approx} \arg\min_{\mathbf{w} \in \mathbb{R}^d} \left\{\frac{1}{2}\|\mathbf{w}^{SGD}(t) - \mathbf{w}\|_2^2 + \lambda_{\gamma,t}\sum_{j=1}^d \bar{s}_j |w_j|\right\}, \quad \text{for any } t > \bar{T}, \tag{8}$$

where $\overset{d}{\approx}$ means "asymptotic in distribution" under the empirical probability measure of the gradients, $\lambda_{\gamma,t} = c\sqrt{\gamma}t^\mu$ and the $\bar{s}_j$ satisfies $\lim_{t\to\infty} |\bar{s}_j - s_j| = 0$ for all $j$.

This theorem holds in the asymptotic regime ($\gamma \to 0$) with a finite time horizon, i.e. any fixed $t \geq \bar{T}$. It is important that $\lambda$ grows with $t$, because the magnitude of SGD asymptotically grows like a Gaussian process, i.e., in $t^{0.5}$. Hence, $\mu$ should be slightly greater than 0.5. The proof of Theorem 3.1 is in Section B.2 of the appendix.

**Remark 3.2** (Condition (A3)). The eigenspace of $\bar{H}$ associated with the zero eigenvalues and $\mathcal{P}_0$ matches when $\mathbf{w}(t)$ and SGD converge to the same flat valley of minima. For the $\ell_2$ loss and in the teacher-student framework, [12, 61, 7] showed $\mathbf{w}(t) \to \mathbf{w}^*$ exponentially fast for one hidden layer networks, so the limit $\bar{H} = H(\mathbf{w}^*)$ and the condition holds. For the cross-entropy loss, we suspect that $\bar{H}$ satisfying (A3) is not a zero matrix, but its exact form needs further investigation.

**Remark 3.3** (Condition (A4)). This condition can be verified (by Problem 3.31 of [57]) if $\operatorname{sign}(\mathbf{w}(t))$ is mainly restricted in the eigenspace of $H(\mathbf{w}(t))$ associated with positive eigenvalues as $t \to \infty$. Empirically, this appears to hold as [25, 21] show that $\mathbf{w}(t)$ lies mainly in the subspace of $H(\mathbf{w}(t))$ associated with the positive eigenvalues, and Figure 2 suggests the angle between $\mathbf{w}(t)$ and $\operatorname{sign}(\mathbf{w}(t))$ is very small.

**Remark 3.4** (Condition (A5)). For (i), under the cross-entropy loss, several papers [56, 24, 34, 41] show that $\mathbf{w}(t)/\|\mathbf{w}(t)\|_2$ converges to a unique direction while $\|\mathbf{w}(t)\|_2 \to \infty$. This implies that $\operatorname{sign}(\mathbf{w}(t))$ stabilizes after a finite time. For the $\ell_2$ loss, [12, 61] show $\mathbf{w}(t) \to \mathbf{w}^*$ for one hidden layer networks under regularity conditions, and the condition follows. The (ii) holds if the learning rate is sufficiently small, so that the deviation between the gradient flow and the SGD is small.

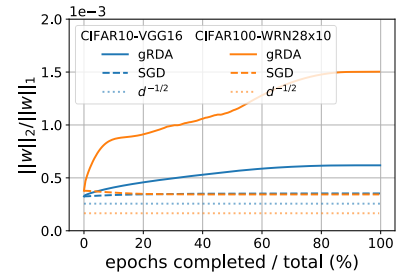

Figure 2: $\|\mathbf{w}\|_2/\|\mathbf{w}\|_1$ is close to its lower bound $d^{-1/2}$ when the coefficients in $\mathbf{w}$ are of similar magnitude, i.e. the direction of $\mathbf{w}$ is the same as $\operatorname{sign}(\mathbf{w})$.

# 4 Empirical experiments

This section presents the empirical performance of (gRDA), and the evidence that (gRDA) performs the directional pruning (Definition 1.1). Section 4.1 considers ResNet50 with ImageNet, and compares with several existing pruning algorithms. To check if (gRDA) performs the directional pruning, Section 4.2 presents the local geometry of the loss around the minimal loss curve that connects the minima found by (SGD) and (gRDA), and Section 4.3 investigates the direction of deviation between the minima found by (SGD) and (gRDA).

## 4.1 ResNet50 on the ImageNet

We use (gRDA) to simultaneously prune and train the ResNet50 [31] on the ImageNet dataset without any post-processing like retraining. The learning rate schedule usually applied jointly with the SGD with momentum does not work well for (gRDA), so we use either a constant learning rate or dropping the learning rate only once in the later training stage. Please find more implementation details in Section C.1 in the appendix. The results are shown in Figure 3, where $\mu$ is the increasing rate of the soft thresholding in the tuning function (5) of (gRDA).

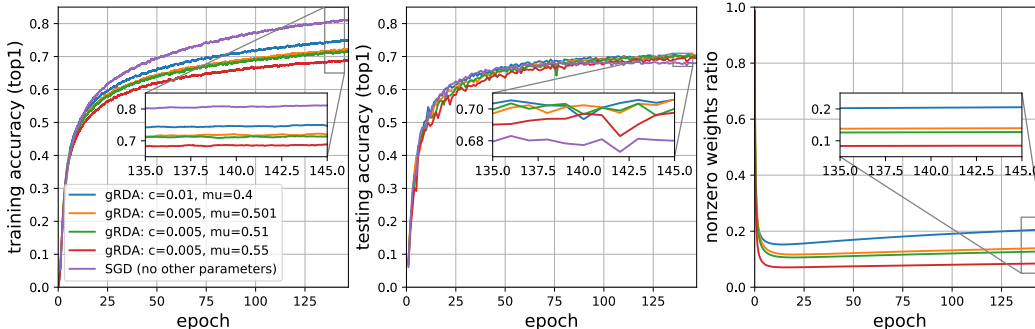

Figure 3: Learning trajectories of (SGD) and (gRDA) for ResNet50 [31] on ImageNet image recognition task. **Left:** top 1 training accuracy. **Center:** top 1 testing accuracy. **Right:** the ratio between the number of nonzero parameters and the total number of parameters. The number of nonzero weights slightly increases, contradicting with Theorem 3.1. This could be because that Assumption (A5) fails due to the large learning rate. $\gamma = 0.1$ for both SGD and gRDA. Minibatch size is 256.

**Accuracy**: gRDAs can perform as accurate as (SGD) after sufficient training. Larger $\mu$ (in the tuning function (5)) can perform worse than (SGD) in the early stage of training, but eventually beat (SGD) in the late stage of training. The training accuracy of (SGD) is higher than that of the gRDAs. This may result from a too large learning rate, so the coefficients $w_j$'s with $s_j < 0$ (in (3)) overshoot and their magnitudes become too large.
**Sparsity**: Sparsity increases rapidly at the early stage of training. With $\mu = 0.55$ in Figure 3, (gRDA) reaches 92% sparsity, while the testing accuracy is higher than (SGD).
**Wall time and memory footprint**: (gRDA) has a slightly higher wall time than (SGD), but the memory footprint is similar. See Section C.5 for a detailed comparison.

The left panel of Figure 4 compares (gRDA) with the magnitude pruning [62] and the variational dropout [42], and (gRDA) is particularly competitive in the high sparsity (90-92%) regime. The right panel of Figure 4 compares different pruning algorithms that do not require expert knowledge for selecting the layerwise pruning level with (gRDA) in terms of the layerwise sparsity. We compare (gRDA) with the Erdős-Rényi-Kernel of [15], variational dropout [42] and a reinforcement-learning based AutoML method [32]. Our (gRDA) achieves a high sparsity 92% with a competitive testing accuracy. In addition, the layerwise sparsity pattern generated by gRDA is similar to the variational dropout and the AutoML, as these methods generate higher sparsity in the 3×3 convolutional layers, and lower sparsity in the 1×1 layers and the initial layers, which are less wide than the latter layers. Among these methods, (gRDA) is unique in that its spirit is interweaving with the local loss landscape.

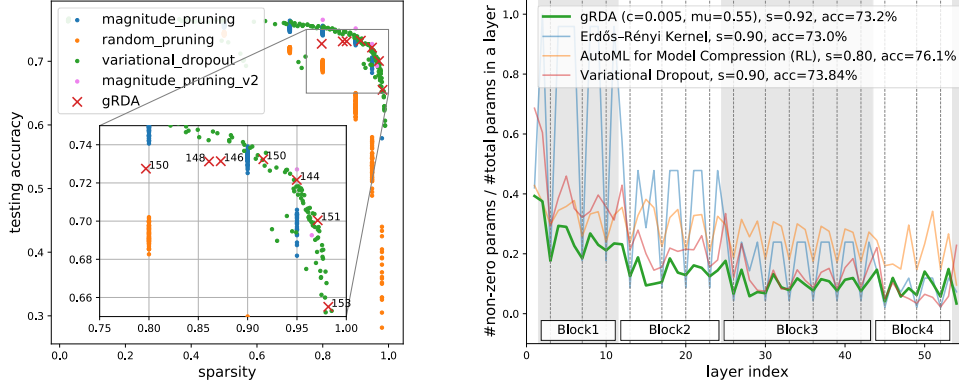

Figure 4: **Left:** A comparison of gRDA with the magnitude pruning [62] and variational dropout [42] with ResNet50 on ImageNet, done by [19] with around 100 epochs using SGD with momentum. Our solution is among the high performers in the very sparse regime (90-92%). The numbers next to the red crosses are the epochs. **Right:** Layerwise sparsity produced by different "automatic" pruning algorithms. All methods show the pattern that the 3x3 conv layers (on dashed lines) are greatly pruned (valleys), and the 1x1 conv layers are less pruned (peaks).

## 4.2  Connectivity between the minimizers of gRDA and SGD

In this section, we check whether (SGD) and (gRDA) reach the same valley, which implies (gRDA) is performing the directional pruning. Similar analysis has been done for the minima found by (SGD) with different initializers [58, 20, 44, 11, 30, 16, 23].

We train VGG16 [55] on CIFAR-10 and WRN28x10 on CIFAR-100 until nearly zero training loss using both (SGD) and (gRDA). The minima here found by (gRDA) generally have sparsity around 90% or higher for larger $\mu$. We use the method of [20] to search for a quadratic Bézier curve of minimal training loss connecting the minima found by the gRDA and SGD, and then visualize the contour of the training losses and testing errors on the hyperplane containing the minimal loss curve. See Sections C.2 and C.3 for details on implementation.

The results are shown for different choices of $\mu$, which is the increasing rate of the soft thresholding in the tuning function (5) of (gRDA). As observed from the contours in Figure 5, the learned parameters of both SGD and gRDA lie in the same valley on the training loss landscape if $\mu$ is properly tuned, namely, $0.6$ for VGG16 and $0.501$ for WRN28x10. This verifies that (gRDA) performs the directional pruning. For large $\mu$, a hill exists on the minimal loss/error path, which may be due to the too large learning rate that leads to large magnitude for the coefficients $j$ with $s_j < 0$. The details (training accuracy, testing accuracy, sparsity) of the endpoints trained on VGG16 and WRN28x10 are shown in Tables 4 and 6 of the Appendix. For the testing error in Figure 5, the gRDA somewhat outperforms SGD when $\mu$ is slightly greater than $0.5$. Interestingly, the neighborhood of the midpoint on the Bézier curve often has a higher testing accuracy than the both endpoints, except for WRN28x10 on CIFAR-100 with $\mu = 0.501$ and $0.55$. This finding resonates with the results of [33].

## 4.3  Direction of $\mathbf{w}^{gRDA} - \mathbf{w}^{SGD}$

The directional pruning (Definition 1.1) implies that the vector $\Delta_n := \mathbf{w}_n^{gRDA} - \mathbf{w}_n^{SGD}$ should lie in $\mathcal{P}_0$ as $n \to \infty$ if tuned appropriately. Unfortunately, checking this empirically requires estimating $\mathcal{P}_0$ which is computationally formidable. Nonetheless, there exists a dominating low dimensional subspace in $\mathcal{P}_0^\perp$ (the subspace orthogonal to $\mathcal{P}_0$); particularly, a few studies [53, 54, 21, 47] have empirically shown that for various networks on the CIFAR-10, the magnitude of the ten leading eigenvalues of $H(\mathbf{w}^{SGD})$ are dominating the others.

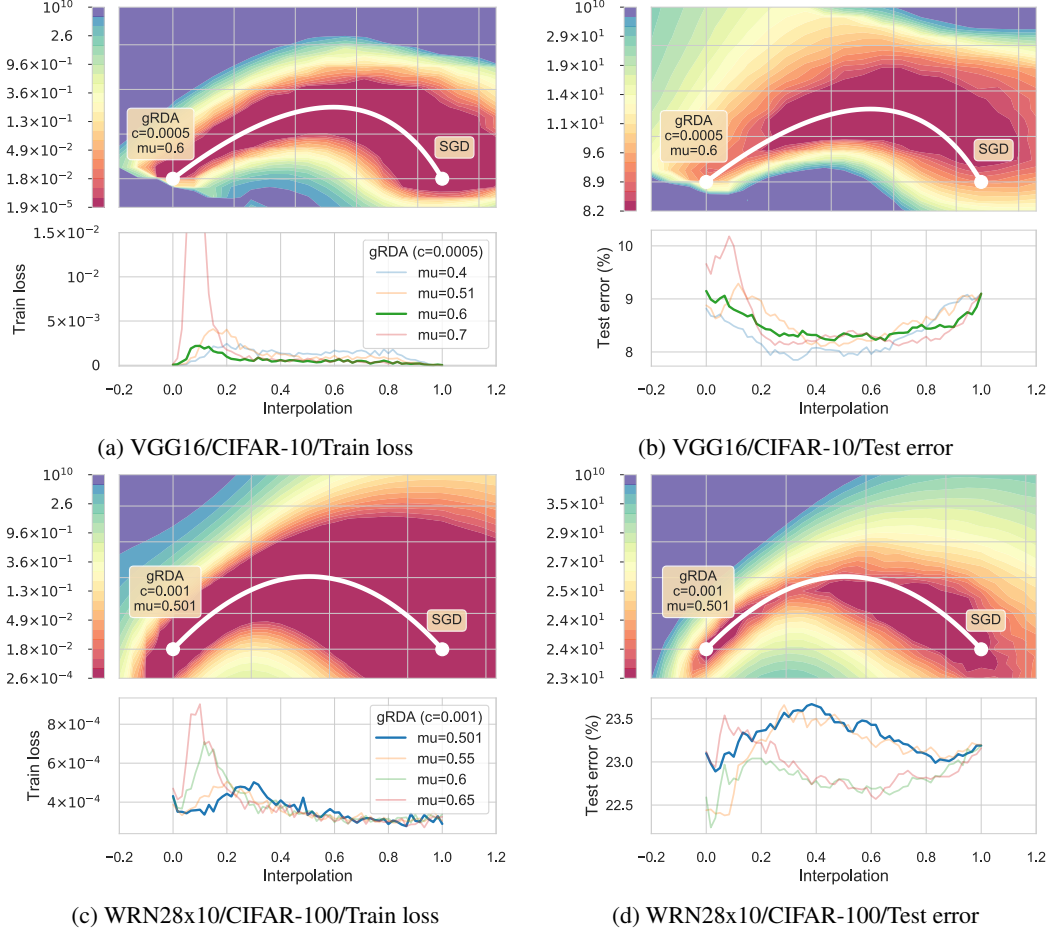

(a) VGG16/CIFAR-10/Train loss

(b) VGG16/CIFAR-10/Test error

(c) WRN28x10/CIFAR-100/Train loss

(d) WRN28x10/CIFAR-100/Test error

Figure 5: The upper figure in each panel shows the contour of training loss and testing error on the hyperplane containing the minimal loss Bézier curve (white) interpolating the minimizers found by the SGD and the gRDA. The lower plot of each panel shows the training loss/testing error on the minimal loss Bézier curve interpolating minimizers of SGD and gRDA under different $\mu$.

Let $\mathcal{P}_n^{top} := \text{span}\{\boldsymbol{u}_{1,n}, \boldsymbol{u}_{2,n}, \ldots, \boldsymbol{u}_{10,n}\}$ be the top subspace spanned by the eigenvectors $\boldsymbol{u}_{j,n}$ associated with the top 10 eigenvalues of $H(\mathbf{w}_n^{SGD})$. Define

$$P_n := \begin{bmatrix} \longleftarrow & \boldsymbol{u}_{1,n} & \longrightarrow \\ \longleftarrow & \boldsymbol{u}_{2,n} & \longrightarrow \\ & \vdots & \\ \longleftarrow & \boldsymbol{u}_{10,n} & \longrightarrow \end{bmatrix}. \tag{9}$$

We train the VGG16 and WRN28x10 on the CIFAR-10, until the training data are nearly interpolated and the training loss is almost zero. During the training process, we fix the initializer and minibatches when we use different optimizers to ensure the comparability. We compute $P_n$ on the training trajectory of VGG16 and WRN28x10. See Section C.4 for details on the computation of these eigenvectors. We test the hypothesis that the proportion of $\Delta_n$ in $\mathcal{P}_n^{top}$ is low, i.e. $\|P_n \Delta_n\|/\|\Delta_n\|$ is low. The results from the VGG16 and WRN28x10 in Figure 6 basically confirm this hypothesis, as the magnitude of the proportion of $\Delta_n$ in $\mathcal{P}_n^{top}$ is very small under the two networks. Particularly, the proportion is always very small for WRN28x10. The results for different $\mu$ are similar, showing that $\Delta_n$ is pointing to the same direction regardless of $\mu$.

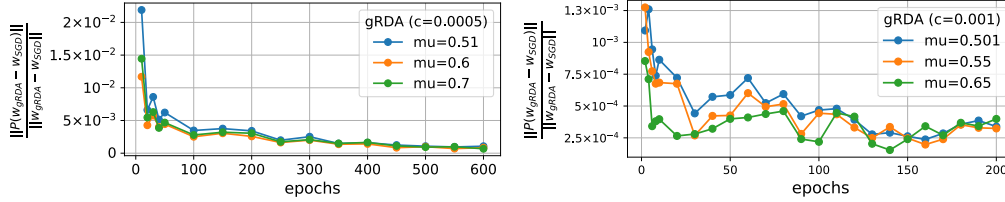

Figure 6: The fraction of the different between SGD and gRDA on the eigenspace associated with the leading 10 eigenvalues. **Left:** VGG16. **Right:** WRN28x10. The $\|\cdot\|$ is the $\ell_2$ norm.

## 5   Discussion and future work

We propose the novel directional pruning for deep neural networks, that aims to prune DNNs while preserving the training accuracy. For implementation, we show that (gRDA) asymptotically achieves the directional pruning after sufficient epochs of training. Empirical evidence shows that our solution yields a accuracy and sparsity tradeoff within the range of many contemporary pruning techniques.

The testing accuracy of (gRDA) is almost always higher than (SGD) if $\mu$ is slightly greater than 0.5 when using the ResNets, and some interpolation between the minima found by (gRDA) and (SGD) often has a better testing accuracy than the two minima; see Figure 5. As suggested by Figure 6, (gRDA) appears to deviate from (SGD) in the flatter directions. These evidences support [30], who argue that the valley of minima is actually asymmetric, and points on the flatter side tend to generalize better. We think a further study of the testing accuracy of (gRDA) along the lines initiated in this work may be an interesting future research topic, as this would shed some light on the mystery of generalization.

## Broader Impact

Our paper belongs to the cluster of works focusing on efficient and resource-aware deep learning. There are numerous positive impacts of these works, including the reduction of memory footprint and computational time, so that deep neural networks can be deployed on devices equipped with less capable computing units, e.g. the microcontroller units. In addition, we help facilitate on-device deep learning, which could replace traditional cloud computation and foster the protection of privacy.

Popularization of deep learning, which our research helps facilitate, may result in some negative societal consequences. For example, the unemployment may increase due to the increased automation enabled by the deep learning.

## Acknowledgments

We thank the anonymous reviewers for the helpful comments. Shih-Kang Chao would like to acknowledge the financial support from the Research Council of the University of Missouri. This work was completed while Guang Cheng was a member of the Institute for Advanced Study, Princeton in the fall of 2019. Guang Cheng would like to acknowledge the hospitality of the IAS and the computational resource it has provided.

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
