[Supplementary Material 1 · 4028_paper_main_and_supplement.pdf]

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

## Footnotes

[2]`https://github.com/pytorch/examples/blob/234bcff4a2d8480f156799e6b9baae06f7ddc96a/`

[4]https://github.com/pytorch/examples/blob/234bcff4a2d8480f156799e6b9baae06f7ddc96a/imagenet/main.py#L400

[5]https://github.com/google-research/google-research/tree/master/state_of_sparsity

[6] https://github.com/timgaripov/dnn-mode-connectivity

[7]`https://github.com/noahgolmant/pytorch-hessian-eigenthings`

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

# APPENDIX

## A  Proof of Remark 1.2

**Proposition A.1.** Consider the optimization problem

$$\arg\min_{w_j \in \mathbb{R}^d} \left\{ f(w_j) := \frac{1}{2}\|w_j^{SGD} - w_j\|_2^2 + \lambda s_j |w_j| \right\}. \tag{A.1}$$

For $w_j^{SGD} \in \mathbb{R}\backslash\{0\}$, $s_j \in \mathbb{R}$, $\lambda > 0$, it has an explicit solution:

$$\widehat{w}_j = \mathrm{sign}(w_j^{SGD})\big[|w_j^{SGD}| - \lambda s_j\big]_+. \tag{A.2}$$

**Proof of Proposition A.1.**
When $s_j = 0$, the solution is $\widehat{w}_j = w_j^{SGD}$.
When $s_j > 0$, the objective function is convex, therefore we only need to verify if $0$ is a subgradient of $f(w_j)$ at $\widehat{w}_j$.

- If $|w_j^{SGD}| > \lambda s_j$, $\widehat{w}_j = w_j^{SGD} - \lambda s_j \,\mathrm{sign}(w_j^{SGD})$. We can see that $\mathrm{sign}(\widehat{w}_j) = \mathrm{sign}(w_j^{SGD})$, and since $\mathrm{sign}(w_j)$ is a subgradient of $|w_j|$, we have $\widehat{w}_j - w_j^{SGD} + \lambda s_j \,\mathrm{sign}(w_j^{SGD}) = 0$ as a subgradient of $f(w_j)$ at $\widehat{w}_j$.
- If $|w_j^{SGD}| \le \lambda s_j$, $\widehat{w}_j = 0$. Since the subgradient set of $|w_j|$ is $[-1, 1]$ at $w_j = 0$, we have $0 \in \big[w_j^{SGD} - \lambda s_j, w_j^{SGD} + \lambda s_j\big] \iff 0 \in \big[\widehat{w}_j - w_j^{SGD} - \lambda s_j, \widehat{w}_j - w_j^{SGD} + \lambda s_j\big]$ (the subgradient set of $f(w_j)$ at $w_j = 0$).

When $s_j < 0$, the objective function is not convex, therefore we need to check the values of $f(w_j)$ at stationary points, boundary points, and non-differentiable points ($w_j = 0$). Since the absolute value function $g(x) = |x|$ is $x$ when $x > 0$ and $-x$ when $x < 0$, we will find the possible stationary points at $w_j > 0$ and $w_j < 0$ separately, and $f(w_j)$ is smooth and strongly convex on each of the two parts.

Without loss of generality, we first assume $w_j^{SGD} > 0$:

- On $w_j > 0$, $f(w_j) = \frac{1}{2}\|w_j^{SGD} - w_j\|_2^2 + \lambda s_j w_j$. The stationary point is $w_j^{SGD} - \lambda s_j$ with objective function value $\frac{(\lambda s_j)^2}{2} + \lambda s_j |w_j^{SGD} - \lambda s_j|$;
- On $w_j < 0$, $f(w_j) = \frac{1}{2}\|w_j^{SGD} - w_j\|_2^2 - \lambda s_j w_j$. The stationary point is $w_j^{SGD} + \lambda s_j$ (if it exists) with objective function value $\widehat{w}_j$ is $\frac{(\lambda s_j)^2}{2} + \lambda s_j |w_j^{SGD} + \lambda s_j|$; note that if $w_j^{SGD} + \lambda s_j \ge 0$, then there is no stationary point in $(-\infty, 0)$;
- At $w_j = 0$, the objective function value is $\frac{(w_j^{SGD})^2}{2}$.

Since $w_j^{SGD} > 0$ and $\lambda s_j < 0$, we have

$$\frac{(\lambda s_j)^2}{2} + \lambda s_j |w_j^{SGD} - \lambda s_j| > \frac{(\lambda s_j)^2}{2} + \lambda s_j |w_j^{SGD} + \lambda s_j|,$$

We also have $\frac{(\lambda s_j)^2}{2} + \lambda s_j |w_j^{SGD} - \lambda s_j| = \lambda s_j w_j^{SGD} - \frac{(\lambda s_j)^2}{2} < 0 < \frac{(w_j^{SGD})^2}{2}$. Therefore the global minimizer of $f(w_j)$ is the right stationary point $\widehat{w}_j = w_j^{SGD} - \lambda s_j = \mathrm{sign}(w_j^{SGD})\max(0, |w_j^{SGD}| - \lambda s_j)$. Similar analysis holds for $w_j^{SGD} < 0$. $\qquad\square$

## B  Proof of theorem

### B.1  Preliminary results

The first result shows that $\mathbf{w}_\gamma(t)$ converges in the functional space $D([0, T])^d$ for any $T > 0$ in probability, where $D([0, T])$ is the space of all functions on $[0, T]$ that are right continuous with left limit. Denote $\xrightarrow{P}$ the convergence in probability. The following result is immediate following by [8].
**Theorem B.1** (Asymptotic trajectory)**.** Suppose (A1) and (A2) hold, and the solution of gradient flow in (GF) is unique, then as $\gamma \to 0$, $\mathbf{w}_\gamma \xrightarrow{P} \mathbf{w}$ in $D([0, T])^d$ for any $T > 0$, where $\mathbf{w}(t)$ is the gradient flow.

The asymptotic trajectory of the dual process $\mathbf{v}_n$ and primal process $\mathbf{w}_n$ are the same, i.e. they are both $\mathbf{w}$. The key reason is that the threshold $g(n,\gamma)$ in $\mathcal{S}_{g(n,\gamma)}(\cdot)$ in (gRDA) tends to zero: $\sup_{t\in[0,T]}\lim_{\gamma\to 0}\big|g(\lfloor t/\gamma\rfloor,\gamma)\big|=0$, so $\mathbf{v}=\mathbf{w}$ in the limit.

**Proof of Theorem B.1.** The proof is an application of Theorem 3.13(a) of [8]. □

The asymptotic trajectory is deterministic, which cannot explain the stochasticity of the learning dynamics. However, in practice, the stochasticity of sampling minibatches has great influence on the quality of training.

We investigate how the stochasticity enters (gRDA). (gRDA) can be written in the stochastic mirror descent (SMD) representation [45]:

$$\mathbf{w}_{n+1} = \mathcal{S}_{g(n,\gamma)}(\mathbf{v}_{n+1}),$$
$$\text{where } \mathbf{v}_{n+1} = \mathbf{v}_n - \gamma\nabla f(\mathbf{w}_n; Z_{n+1}) \tag{SMD}$$

The process $\mathbf{v}_n$ is an auxiliary process in the dual space (generated by the gradients), and the primal process $\mathbf{w}_n$, corresponding to the parameters of DNN, can be represented as a transformation of the dual process by $\mathcal{S}_{g(n,\gamma)}$.

Random gradients enter $\mathbf{v}_n$, while $\mathbf{w}_n$ is obtained by taking a deterministic transformation of $\mathbf{v}_n$. To characterize the randomness of $\mathbf{v}_n$, consider $\mathbf{v}_\gamma(t) := \mathbf{v}_{\lfloor t/\gamma\rfloor}$ the piecewise constant interpolated process, where $\lfloor a\rfloor$ takes the greatest integer that is less than or equal to $a$. This next theorem provides the distribution of $\mathbf{v}_\gamma(t)$.

**Theorem B.2** (Distributional dynamics). Suppose (A1), (A2) and (A3) hold. In addition, suppose the root of the coordinates in $\mathbf{w}(t)$ occur at time $\{T_k\}_{k=1}^\infty \subset [0,\infty)$. Let $\mathbf{w}_0$ with $w_{0,j}\neq 0$ (e.g. from a normal distribution) and $T_0=0$. Then, as $\gamma$ is small, for $t\in(T_K,T_{K+1})$,

$$\mathbf{v}_\gamma(t) \overset{d}{\approx} \boldsymbol{g}^\dagger(t) + \mathbf{w}(t) - \sqrt{\gamma}\boldsymbol{\delta}(t) + \sqrt{\gamma}\int_0^t \Phi(t,s)^\top \Sigma^{1/2}(\mathbf{w}(s))d\mathbf{B}(s), \tag{B.1}$$

where $\overset{d}{\approx}$ denotes approximately in distribution, $\Sigma(\mathbf{w})$ is the covariance kernel defined in (7) and $\mathbf{B}(s)$ is a $d$-dimensional standard Brownian motion, and

$$\boldsymbol{g}^\dagger(t) := \sqrt{\gamma}ct^\mu \operatorname{sign}(\mathbf{w}(T_K^+)) \tag{B.2}$$

$$\boldsymbol{\delta}(t) := c\mu\int_0^t s^{\mu-1}\Phi(t,s)\operatorname{sign}(\mathbf{w}(s))ds + c\sum_{k=1}^K \Big\{\Phi(t,T_k)\big\{\operatorname{sign}(\mathbf{w}(T_k^+)) - \operatorname{sign}(\mathbf{w}(T_k^-))\big\}T_k^\mu\Big\} \tag{B.3}$$

$\Phi(t,s)\in\mathbb{R}^{d\times d}$ is the principal matrix solution (Chapter 3.4 of [57]) of the ODE system:

$$d\mathbf{x}(t) = -H(\mathbf{w}(t))\mathbf{x}(t)dt, \quad \mathbf{x}(0)=\mathbf{x}_0. \tag{B.4}$$

The proof follows by a functional central limit theorem in [8] for Markov processes generated by regularized stochastic algorithms.

**Proof of Theorem B.2.** Consider the centered and scaled processes

$$\mathbf{V}_\gamma(t) := \frac{\mathbf{v}_\gamma(t)-\mathbf{w}(t)}{\sqrt{\gamma}}. \tag{B.5}$$

By Theorem 3.13(b) of [8], $\mathbf{V}_\gamma \overset{d}{\approx} \mathbf{V}$ on $(T_k,T_{k+1})$ for each $k=0,\dots,K$ as $\gamma$ is small, where $\mathbf{V}$ obeys the stochastic differential equation (SDE):

$$d\mathbf{V}(t) = -H(\mathbf{w}(t))\big[\mathbf{V}(t) - \operatorname{sign}(\mathbf{w}(t))ct^\mu\big]dt + \Sigma^{1/2}(\mathbf{w}(t))d\mathbf{B}(t), \tag{B.6}$$

with initial $\mathbf{V}(T_k)=\mathbf{V}(T_k^-)$, and $\mathbf{B}(t)$ is the $d$-dimensional standard Brownian motion. Note that $\mathbf{V}(T_0)=\mathbf{V}(0)=\mathbf{V}_\gamma(0)=0$ almost surely.

Under condition (A3), the global Lipschitz and linear growth conditions hold, so there exists a unique strong solution of (B.6) by Theorem 5.2.9 of [35].

In addition, by condition (A3), the solution operator $\Phi(t,s)$ of the inhomogeneous ODE system,
$$d\mathbf{x}(t) = -H(\mathbf{w}(t))\mathbf{x}(t)dt, \quad \mathbf{x}(s) = \mathbf{x}_s, \tag{B.7}$$
uniquely exists, and the solution is $\mathbf{x}(t) = \Phi(t,s)\mathbf{x}_s$ by Theorem 5.1 of [50]. $\Phi(t,s)$ satisfies the properties in Theorem 5.2 of [50]; in particular, for all $0 < s < r < t$,
$$(s,t) \mapsto \Phi(t,s) \text{ is continuous} \tag{B.8}$$
$$\Phi(t,t) = I_d \tag{B.9}$$
$$\frac{\partial}{\partial t}\Phi(t,s) = -H(\mathbf{w}(t))\Phi(t,s), \tag{B.10}$$
$$\frac{\partial}{\partial s}\Phi(t,s) = \Phi(t,s)H(\mathbf{w}(s)), \tag{B.11}$$
$$\Phi(t,s) = \Phi(t,r)\Phi(r,s). \tag{B.12}$$
Recall from (B.6) that for $t \in (T_k, T_{k+1})$,
$$d\mathbf{V}(t) = -H(\mathbf{w}(t))\big[\mathbf{V}(t) - \mathrm{sign}(\mathbf{w}(t))ct^\mu\big]dt + \Sigma^{1/2}(\mathbf{w}(t))d\mathbf{B}(t), \tag{B.13}$$
with initial distribution $\mathbf{V}(T_k^-)$. It can be verified by (B.10) and Ito calculus that for $t \in (T_k, T_{k+1})$,
$$\mathbf{V}(t) = \Phi(t,T_k)\mathbf{V}(T_k^-) + \int_{T_k}^t \Phi(t,s)H(\mathbf{w}(s))\,\mathrm{sign}(\mathbf{w}(s))cs^\mu ds + \int_{T_k}^t \Phi(t,s)\Sigma^{1/2}(\mathbf{w}(s))d\mathbf{B}(s)$$
$$\tag{B.14}$$
is the solution of (B.13).

Integration by part, (B.11) and (B.9) yield that
$$\int_{T_k}^t \Phi(t,s)H(\mathbf{w}(s))\,\mathrm{sign}(\mathbf{w}(s))cs^\mu ds$$
$$= ct^\mu\,\mathrm{sign}(\mathbf{w}(t)) - cT_k^\mu\Phi(t,T_k^+)\,\mathrm{sign}(\mathbf{w}(T_k^+)) - \int_{T_k}^t \Phi(t,s)c\mu s^{\mu-1}\,\mathrm{sign}(\mathbf{w}(s))ds.$$
If $t > T_K$, last display, induction, (B.12) and (B.14) imply
$$\mathbf{V}(t)$$
$$= \Phi(t,T_K)\mathbf{V}(T_K^-) + \int_{T_K}^t \Phi(t,s)H(\mathbf{w}(s))\,\mathrm{sign}(\mathbf{w}(s))cs^\mu ds + \int_{T_K}^t \Phi(t,s)\Sigma^{1/2}(\mathbf{w}(s))d\mathbf{B}(s)$$
$$= ct^\mu\,\mathrm{sign}(\mathbf{w}(T_K^+)) + \Phi(t,T_K)\mathbf{V}(T_K^-) - cT_K^\mu\Phi(t,T_K)\,\mathrm{sign}(\mathbf{w}(T_K^+))$$
$$\quad - \int_{T_K}^t \Phi(t,s)c\mu s^{\mu-1}\,\mathrm{sign}(\mathbf{w}(s))ds + \int_{T_K}^t \Phi(t,s)\Sigma^{1/2}(\mathbf{w}(s))d\mathbf{B}(s)$$
$$= ct^\mu\,\mathrm{sign}(\mathbf{w}(T_K^+)) + \Phi(t,T_{K-1})\mathbf{V}(T_{K-1}^-) - cT_K^\mu\Phi(t,T_K)\{\mathrm{sign}(\mathbf{w}(T_K^+)) - \mathrm{sign}(\mathbf{w}(T_K^-))\}$$
$$\quad - cT_{K-1}^\mu\Phi(t,T_{K-1})\,\mathrm{sign}(\mathbf{w}(T_{K-1}^+)) - \int_{T_{K-1}}^t \Phi(t,s)c\mu s^{\mu-1}\,\mathrm{sign}(\mathbf{w}(s))ds$$
$$\quad + \int_{T_{K-1}}^t \Phi(t,s)\Sigma^{1/2}(\mathbf{w}(s))d\mathbf{B}(s)$$
$$\vdots$$
$$= ct^\mu\,\mathrm{sign}(\mathbf{w}(T_K^+)) + \Phi(t,T_{K-1})\underbrace{\mathbf{V}(0)}_{=\,0\text{ a.s.}} - \boldsymbol{\delta}(t) + \int_0^t \Phi(t,s)\Sigma^{1/2}(\mathbf{w}(s))d\mathbf{B}(s),$$
where $\boldsymbol{\delta}(t) = \boldsymbol{\delta}_1(t) + \boldsymbol{\delta}_2(t)$ with
$$\boldsymbol{\delta}_1(t) := c\mu \int_0^t s^{\mu-1}\Phi(t,s)\,\mathrm{sign}(\mathbf{w}(s))ds,$$
$$\tag{B.15}$$
$$\boldsymbol{\delta}_2(t) := c\sum_{k=1}^K \Big\{\Phi(t,T_k)\big\{\,\mathrm{sign}(\mathbf{w}(T_k^+)) - \mathrm{sign}(\mathbf{w}(T_k^-))\big\}T_k^\mu\Big\}.$$
$$\square$$

## B.2 Proof of Theorem 3.1

By virtue of Remark 1.2, it is enough to show that

$$w_{\gamma,j}(t) \stackrel{d}{\approx} \text{sign}\{w_j^{SGD}(t)\}\{|w_j^{SGD}(t)| - \lambda_{\gamma,t}\bar{s}_j\}_+ \tag{B.16}$$

where $\lambda_{\gamma,t} = c\sqrt{\gamma}t^\mu$, and $\bar{s}_j = s_j + o(1)$. This is implied by the following theorem.

**Theorem B.3.** Suppose (A1)-(A5) hold. Assume that $\mu \in (0.5, 1)$ and $c > 0$ in (5). In addition, if $\text{sign}\{w_j(t)\} = \text{sign}\{w_j^{SGD}(t)\}$ for all $j = 1, \ldots, d$, then, for a sufficiently large $t > \bar{T}$,

$$w_{\gamma,j}(t) \stackrel{d}{\approx} \text{sign}\{w_j^{SGD}(t)\}\{|w_j^{SGD}(t)| - \text{sign}\{w_j^{SGD}(t)\}\sqrt{\gamma}\delta_j(t)\}_+ \tag{B.17}$$

where $\boldsymbol{\delta}(t)$ has an explicit form in (B.15) in the appendix, and satisfies as $t \to \infty$,

$$\boldsymbol{\delta}(t) = ct^\mu \Pi_0 \text{sign}(\mathbf{w}(t)) + o(t^\mu) + O(t^{\mu-1}), \tag{B.18}$$

and $\Pi_0$ is the orthogonal projection on the eigenspace corresponding to zero eigenvalues of $\bar{H}$.

The proof of (B.17) will be based on Theorem B.2, and (B.18) relies on the Levinson theorem, which provides asymptotic solution of the ODE in (6).

**Proof of Theorem B.3.** From (B.1),

$$\mathbf{v}_\gamma(t) \stackrel{d}{\approx} \boldsymbol{g}^\dagger(t) + \mathbf{w}(t) - \sqrt{\gamma}\boldsymbol{\delta}(t) + \sqrt{\gamma}\boldsymbol{U}(t).$$

where we recall $\boldsymbol{g}^\dagger(t) := \sqrt{\gamma}ct^\mu \text{sign}(\mathbf{w}(T_K^+))$ and $\boldsymbol{\delta}(t)$ in (B.3).

$$\boldsymbol{U}(t) := \int_0^t \Phi(t,s)^\top \Sigma^{1/2}(\mathbf{w}(s))d\mathbf{B}(s). \tag{B.19}$$

**Step 1 Show** $\text{sign}\{\mathbf{v}_\gamma(t)\} = \text{sign}\{\boldsymbol{g}^\dagger(t)\}$ **with high probability.**
The goal of this step is achieved if we show

$$|\boldsymbol{g}^\dagger(t)| > \sqrt{\gamma}|-\boldsymbol{\delta}(t) + \boldsymbol{U}(t)| \quad (|\cdot| \text{ and } > \text{ are componentwise}). \tag{B.20}$$

To this end, we will show that

$$|\boldsymbol{g}^\dagger(t)| - \sqrt{\gamma}|\boldsymbol{\delta}(t)| > \sqrt{\gamma}|\boldsymbol{U}(t)|. \tag{B.21}$$

Clearly, this implies (B.20).
Recall that $\boldsymbol{\delta}(t) = \boldsymbol{\delta}_1(t) + \boldsymbol{\delta}_2(t)$ where $\boldsymbol{\delta}_j$'s are defined in (B.15). By (A4),

$$|g_j^\dagger(t)| - |\delta_{1,j}(t)| \geq c_1 t^\mu, \tag{B.22}$$

for some $c_1 > 0$. On the other hand, $\delta_{2,j}(t)$ is defined in (B.15). By (A4) and (A5),

$$|\delta_{2,j}(t)| < C\bar{T}^\mu.$$

Hence, $|\delta_{2,j}(t)| = O(1)$.
From above, we get

$$|\boldsymbol{g}^\dagger(t)| - \sqrt{\gamma}|\boldsymbol{\delta}(t)| > c_0 t^\mu.$$

Using the fact that $\boldsymbol{U}(t)$ as a Gaussian process grows like $t^{1/2}$ up to multiplicative logarithmic terms with high probability (Theorem D.4 of [51]), $\boldsymbol{U}(t)$ is dominated by $c_0 t^\mu$ with $\mu > 0.5$, the proof of (B.21) is complete.
**Step 2 Proof of** (B.17).
By the formulation (gRDA), $\mathbf{w}_\gamma(t) = \mathcal{S}_{\boldsymbol{g}^\dagger(t)}(\mathbf{v}_\gamma(t))$. Hence, for $j \leq d$, note that $\boldsymbol{g}^\dagger(t) = (g_1^\dagger(t), \ldots, g_d^\dagger(t))$, and the piecewise constant process of SGD (with the same minibatch sequence as gRDA):

$$w_{\gamma,j}^{SGD}(t) \stackrel{d}{\approx} w_j(t) + \sqrt{\gamma}U_j(t), \tag{B.23}$$

which can be obtained under the same assumptions in this Theorem by [6, 5].

Hence,

$$w_{\gamma,j}(t) \overset{d}{\approx} \mathcal{S}_{g_j^\dagger(t)}\{v_{\gamma,j}(t)\}$$

$$\overset{d}{\approx} \begin{cases} w_{\gamma,j}^{SGD}(t) - \sqrt{\gamma}\delta_j(t), & \text{if } \operatorname{sign}(w_j(t))\{w_j(t) - \sqrt{\gamma}\delta_j(t) + \sqrt{\gamma}U_j(t)\} > 0, \\ 0, & \text{otherwise.} \end{cases}$$

For (B.17), if $\operatorname{sign}\{w_j(t)\} = \operatorname{sign}\{w_j^{SGD}(t)\}$, $\operatorname{sign}(w_j(t))\{w_j(t) - \sqrt{\gamma}\delta_j(t) + \sqrt{\gamma}U_j(t)\} > 0$ can be rewritten by using (B.23) that

$$|w_{\gamma,j}^{SGD}(t)| > \operatorname{sign}(w_j^{SGD}(t))\sqrt{\gamma}\delta_j(t).$$

Thus, (B.17) follows.

**Step 3  Proof of** (B.18). The proof relies on the Levinson theorem [14] from the theory of asymptotic solution of ordinary differential equations. Note that $\bar{H}$ as a real symmetric matrix is diagonalizable, i.e. there exists orthonormal matrix $P$ and diagonal matrix $\Lambda$ with non-negative values such that $\bar{H} = P\Lambda P^\top$, where $\Lambda = \operatorname{diag}(\lambda_1, \ldots, \lambda_d)$, and the column vectors of $P$ are eigenvectors $\boldsymbol{u}_j$. Let $a_t \to 0$ satisfying

$$\int_t^\infty \|H(\mathbf{w}(s)) - \bar{H}\|ds = O(a_t).$$

The Levinson theorem (Theorem 1.8.1 on page 34 of [14]), together with the estimation of the remainder term on page 15-16 of [14], imply that the principal matrix solution $\Phi(t,s)$ in (B.7) satisfies

$$\Phi(\tau,s) = P(I_d + O(a_\tau))e^{-\Lambda(\tau-s)}P^\top = P_0 P_0^\top + O(e^{-\underline{\lambda}(\tau-s)}) + O(a_\tau), \tag{B.24}$$

where $\underline{\lambda}$ is the least positive eigenvalue of $\bar{H}$, the column vectors of $P_0$ are eigenvectors associated with the zero eigenvalue. Clearly, $P_0 P_0^\top = \sum_{j:\bar{H}\boldsymbol{u}_j=0} \boldsymbol{u}_j \boldsymbol{u}_j^\top$.

Recall the time $\{T_k\}_{k=1}^\infty$ defined in Theorem B.2. By the condition of this Proposition, there exists $K \in \mathbb{N}$ such that $\operatorname{sign}(\mathbf{w}(t)) = \operatorname{sign}(\mathbf{w}(T_K))$ for all $t > T_K$. Recall that $\boldsymbol{\delta}(t) = \boldsymbol{\delta}_1(t) + \boldsymbol{\delta}_2(t)$ where $\boldsymbol{\delta}_1(t)$ and $\boldsymbol{\delta}_2(t)$ are defined in (B.15). Then

$$\boldsymbol{\delta}_2(t) = cP_0 P_0^\top \sum_{k=1}^K \{\operatorname{sign}(\mathbf{w}(T_k^+)) - \operatorname{sign}(\mathbf{w}(T_k^-))\}T_k^\mu + O(e^{-\underline{\lambda}(t-T_K)}T_K^\mu) + O(a_t T_K^\mu)$$

$$= -c\mu P_0 P_0^\top \left(\int_0^{T_K} s^{\mu-1}\operatorname{sign}(\mathbf{w}(s))ds - T_K^\mu \operatorname{sign}(\mathbf{w}(T_K^+))\right) + O(e^{-\underline{\lambda}(t-T_K)}T_K^\mu) + O(a_t T_K^\mu). \tag{B.25}$$

On the other hand, inputing (B.24) into $\boldsymbol{\delta}_1$,

$$\boldsymbol{\delta}_1(t) = c\mu \int_0^t s^{\mu-1}\Phi(t,s)\operatorname{sign}(\mathbf{w}(s))ds$$

$$= c\mu P_0 P_0^\top \int_0^t s^{\mu-1}\operatorname{sign}(\mathbf{w}(s))ds + I(t) + II(t), \tag{B.26}$$

and note that

$$I(t) \lesssim \int_0^t s^{\mu-1}e^{-\underline{\lambda}(t-s)}\|\operatorname{sign}(\mathbf{w}(s))\|ds \leq d^{1/2}\int_0^t s^{\mu-1}e^{-\underline{\lambda}(t-s)}ds = O(t^{\mu-1}),$$

$$II(t) \lesssim \int_0^t s^{\mu-1}a_t\operatorname{sign}(\mathbf{w}(s))ds = O(t^\mu a_t),$$

where the bound of $I$ is obtained by using similar arguments as the proof of Theorem 4.2 of [8] provided that $\mu < 1$. The bound for $II(t)$ is elementary.

Note that $t^\mu a_t > T_K^\mu a_t$ by $\mu > 0$ and $t > T_K$, and that $e^{-\underline{\lambda}(t-T_K)} \to 0$ exponentially in $t$ as $T_K$ is fixed. Combining (B.25) and (B.26) yield

$$\boldsymbol{\delta}_1(t) + \boldsymbol{\delta}_2(t) = ct^\mu P_0 P_0^\top \operatorname{sign}(\mathbf{w}(t)) + O\big(\max\{t^\mu a_t, t^{\mu-1}\}\big),$$

where $P_0 P_0^\top$ is a projection matrix projecting on the subspace spanned by the eigenvectors corresponding to zero eigenvalues. Set $\Pi_0 = P_0 P_0^\top$.

$\square$

## C    Algorithms for implementation

### C.1    Basic version with a constant learning rate

---

**Algorithm 1:** Generalized Regularized Dual Averaging (gRDA) with $\ell_1$ penalty

---

**Hyperparameters**: $\gamma$: learning rate
**Hyperparameters**: $c \in [0, \infty], \mu \in (0.5, 1)$: fixed parameters in $g(n, \gamma) = c\gamma^{1/2}(n\gamma)^{\mu}$
**Initialization**        : $n \leftarrow 0$: iteration number
**Initialization**        : $\mathbf{w}_0$: initial parameters
**Initialization**        : $G_0 \leftarrow \mathbf{w}_0$: accumulator of gradients
**while** *Testing accuracy not converged* **do**
  $\quad n \leftarrow n + 1$;
  $\quad G_n \leftarrow G_{n-1} + \gamma \nabla f_j(\mathbf{w}_{n-1}; Z_n)$;
  $\quad \mathbf{w}_n \leftarrow \mathrm{sign}(G_n) \max(0, |G_n| - g(n, \gamma))$ ;        `// entry-wise soft-thresholding`
**end**

---

### C.2    Modified tuning function for constant-and-drop learning rate

In practice, a "constant-and-drop" learning rate schedule is usually adopted. For example, the default learning rate schedule in the PyTorch implementation of ResNet on ImageNet is divided by 10 folds for every 30 epochs.[2] In this case, we replace Algorithm 1 by Algorithm 2 below, where we set the solf-thresholding level $\widetilde{g}(n)$ that accumulates the increments of $g(n, \gamma)$ at every iteration.

---

**Algorithm 2:** gRDA with constant-and-drop learning rates

---

**Hyperparameters**: $\{\gamma_n\}$: learning rate schedule
**Hyperparameters**: $c \in [0, \infty], \mu \in (0.5, 1)$: fixed parameters in $g(n, \gamma) = c\gamma^{1/2}(n\gamma)^{\mu}$
**Initialization**        : $n \leftarrow 0$: iteration number
**Initialization**        : $\mathbf{w}_0$: initial parameters
**Initialization**        : $G_0 \leftarrow \mathbf{w}_0$: accumulator of gradients
**Initialization**        : $\widetilde{g}(0) \leftarrow 0$: accumulator of thresholds
**while** *Testing accuracy not converged* **do**
  $\quad n \leftarrow n + 1$;
  $\quad G_n \leftarrow G_{n-1} + \gamma_n \nabla f_j(\mathbf{w}_{n-1}; Z_n)$;
  $\quad \widetilde{g}(n) \leftarrow \widetilde{g}(n-1) + (g(n, \gamma_n) - g(n-1, \gamma_n))$ ;   `// threshold increment for` $\gamma_n$
  $\quad \mathbf{w}_n \leftarrow \mathrm{sign}(G_n) \max(0, |G_n| - \widetilde{g}(n))$ ;
**end**

---

## D    Details on numerical analysis

We did all experiments in this paper using servers with 2 GPUs (Nvidia Tesla P100 or V100, 16GB memory), 2 CPUs (each with 12 cores, Intel Xeon Gold 6126), and 192 GB memory. We use PyTorch [49] for all experiments.

### D.1    Details for experiments on ImageNet

We use the codes from PyTorch official implementation[3] of training ResNet-50 on ImageNet. The batch size used in all ImageNet experiments is 256 (the default value for training ResNet-50) and the data preprocessing module in the original codes is used as well. We follow the separation of training and validation dataset in the official setting of `ILSVRC2012` task (1281167 images in training and 50000 images in validation).

`imagenet/main.py#L400`
[3]`https://github.com/pytorch/examples/blob/234bcff4a2d8480f156799e6b9baae06f7ddc96a/`
`imagenet/main.py`

Figure 3 presents the training accuracy, testing accuracy as well as sparsity. Note that the state-of-the-art performance of ResNet50 on ImageNet (top-1 accuracy 77.15% [31]) using the SGD with momentum and weight decay is higher than the basic SGD (top-1 accuracy around 68% as shown in Figure 3). This is because we fix the learning rate at 0.1, and run SGD without momentum or weight decay. Compared with the SGD, gRDA has a lower training accuracy but a slightly higher testing accuracy. When we increase $\mu$, the training accuracy decreases since larger $\mu$ induces higher sparsity. However, the testing accuracy for all choices of $\mu$'s in gRDA are similar.

Figure 7: A comparison of gRDA with the magnitude pruning [62] and variational dropout [42] with ResNet50 on ImageNet. The numbers next to the red crosses are the epochs.

Table 1: The parameters for gRDA in Figure 7.

| $c$ | $\mu$ | Epoch | Sparsity (%) | Test Acc. (%) |
|---|---|---|---|---|
| PyTorch Official Learning Rate | | | | |
| 0.005 | 0.85 | 85 | 99.84 | 22.69 |
| 0.005 | 0.8 | 90 | 99.51 | 43.47 |
| 0.005 | 0.65 | 91 | 97.05 | 66.46 |
| 0.005 | 0.75 | 92 | 98.99 | 56.15 |
| 0.005 | 0.7 | 94 | 98.26 | 62.60 |
| 0.004 | 0.7 | 95 | 97.69 | 65.17 |
| 0.003 | 0.7 | 95 | 96.87 | 67.28 |
| 0.004 | 0.65 | 95 | 96.36 | 68.06 |
| 0.003 | 0.75 | 103 | 98.10 | 63.76 |
| 0.002 | 0.75 | 105 | 97.06 | 67.23 |
| 0.004 | 0.75 | 121 | 98.62 | 60.12 |
| Only Drop at Epoch 140 | | | | |
| 0.005 | 0.6 | 144 | 94.98 | 72.16 |
| 0.005 | 0.51 | 146 | 87.28 | 73.14 |
| 0.005 | 0.501 | 148 | 86.09 | 73.13 |
| 0.005 | 0.55 | 150 | 91.60 | 73.24 |
| 0.01 | 0.4 | 150 | 79.69 | 72.75 |
| 0.005 | 0.65 | 151 | 97.10 | 70.04 |
| 0.005 | 0.7 | 153 | 98.17 | 65.51 |

The left panel of Figure 4 is reproduced from the bottom panel of Figure 3 in [19], and we add the results of gRDA which are marked by the red crosses. The gRDA is performed using a "constant-and-drop" learning rate schedule. Concretely, $\gamma = 0.1$ for epoch 1 to 140, and $\gamma = 0.01$ for epoch after 140. Figure 7 provides additional results of the gRDA using the learning rate schedule given in the PyTorch official implementation:[4]

- $\gamma = 0.1$ for epoch 1 to 30
- $\gamma = 0.01$ for epoch 31 to 60
- $\gamma = 0.001$ for epoch 61 to 90, and $\gamma = 0.0001$ for epoch after 90

We found that the gRDA relatively underperforms with this learning rate schedule. This schedule for the ImageNet is usually applied jointly with the SGD with Polyak's momentum. As we find that SGD without momentum only yields a test accuracy of 68.76% for ImageNet-ResNet50 under this learning rate schedule, we suspect that the absence of momentum in (gRDA) could be a reason for the underperformance.

The right panel of Figure 4 shows the layerwise sparsity using several different pruning methods. The results of AutoML for Model Compression are from stage4 in Figure 3 of [32]. And the results of Variational Dropout are from [19][5] and we choose the one with 90% sparsity. The results of Erdős-Rényi-Kernel are from Figure 12 (90 % Sparse ERK, i.e. the subfigure on right) in [15].

## D.2 Settings of training models on CIFAR-10 and CIFAR-100

The two datasets CIFAR-10 and CIFAR-100 are described in [36]. Particularly, we follow the separation of training and validation dataset in the official setting (50000 images in training and 10000 images in validation for both CIFAR-10 and CIFAR-100). For our experiments on CIFAR-10 and CIFAR-100, we mostly follow the codes of [20].[6] The batch size used in all experiments is 128 and the data preprocessing module in the original codes is used as well. We follow the example in [20] and set `--use_test`. For optimizers, we use `SGD(momentum=0,weight_decay=0)` and `gRDA(`$c,\mu$`)` as defaults. For the two architectures we used, VGG16, as in its vanilla version, does not have batch normalization, while WRN28x10 has batch normalization.

For both SGD and gRDA, the base learning rate $\gamma$ and epochs are the same as summarized in Table 2. We follow the learning rate schedule adopted by [20]:

- For the first 50% of epochs, we use the base learning rate, i.e. $\gamma_i = \gamma$, if $\frac{i}{n} \in [0, 0.5)$;
- For 50% to 90% of epochs, the learning rate decreases linearly from the base learning rate to 1% of the base learning rate, i.e. $\gamma_i = (1.0 - (\frac{i}{n} - 0.5)\frac{0.99}{0.4})\gamma$, if $\frac{i}{n} \in [0.5, 0.9)$;
- For the last 10% of epochs, we keep using the 1% of the base learning rate as learning rate, i.e. $\gamma_i = 0.01\gamma$, if $\frac{i}{n} \in [0.9, 1]$.

Table 2: Details for training models on CIFAR-10 and CIFAR-100. The minibatch size is 128. Parameters not included in this table are selected as the default values in the code of [20].

| Data | Model | Base Learning Rate | Epochs | Results | Used in Section D.3 (connectivity) | Used in Section D.4 (projection) |
|---|---|---|---|---|---|---|
| CIFAR-10 | VGG16 | 0.1 | 600 | Figure 8 Table 4 | Yes | Yes |
| CIFAR-10 | WRN28x10 | 0.1 | 200 | Figure 9 Table 5 | No | Yes |
| CIFAR-100 | WRN28x10 | 0.1 | 200 | Figure 10 Table 6 | Yes | No |

We train our models with ten different seeds using both SGD and gRDA, and show the training accuracy/loss, testing accuracy/loss, and sparsity along the training process in Figure 8, 9, and 10 (as in Figure 3). Table 4, 5 and 6 provide specific numbers for selected epochs.

For Figure 2, we show the result of the first seed under the two settings: VGG16-CIFAR-10 (gRDA with $c = 0.0005, \mu = 0.51$) and WRN28x10-CIFAR-100 (gRDA with $c = 0.001, \mu = 0.501$). We also select other seeds among the ten seeds, and the curve nearly overlaps with each other. Therefore we only show the result of the first seed.

## D.3 Details for Section 4.2

For the analysis of the connectivity between two neural networks, we follow [20] to train a quadratic Bézier curve interpolating two fixed endpoints $\widehat{\mathbf{w}}_1$ and $\widehat{\mathbf{w}}_2$, which are parameters trained by the SGD and the gRDA, respectively. $\widehat{\mathbf{w}}_1$ and $\widehat{\mathbf{w}}_2$ are trained with 600 epochs for VGG16, and 200 epochs for WRN28x10. Instead of training the entire curve, we follow [20] and train random points sampled from the curve between the two endpoints, i.e., we sample $t \sim \text{Uniform}(0, 1)$ and generate a model with weights being $\theta_{\mathbf{w}}(t) = \widehat{\mathbf{w}}_1(1-t)^2 + \widehat{\mathbf{w}}_2 t^2 + 2t(1-t)\mathbf{w}$ with a trainable vector $\mathbf{w}$ (initialized at $(\widehat{\mathbf{w}}_1 + \widehat{\mathbf{w}}_2)/2$), and train $\mathbf{w}$ with the loss $\ell(\theta_{\mathbf{w}}(t))$ at a fixed $t$ using the SGD to get $\widehat{\mathbf{w}}_3$.

We use the program in [20] to produce Figure 5, and the settings are summarized in Table 3. Parameters that are more technical are set by the default values in the GitHub repository of [20]. The top panels of Figure 5 illustrate the training loss contour on the hyperplane determined by the $(\widehat{\mathbf{w}}_1, \widehat{\mathbf{w}}_2, \widehat{\mathbf{w}}_3)$. The bottom panels are obtained through the models on the curve, i.e. the model $\theta_{\widehat{\mathbf{w}}_3}(t)$ for $t \in [0, 1]$. More results are showing in Figure 11.

Table 3: Details for training quadratic Bézier curve on models with CIFAR-10 and CIFAR-100. Here, we use the SGD with momentum in the CIFAR-10 task because the SGD without momentum results in NaN during training. Parameters not included in this table are selected as the default values in the code of [20].

| Data | Model | Learning Rate | Epochs | Momentum | Weight Decay |
|------|-------|---------------|--------|----------|--------------|
| CIFAR-10 | VGG16 | 0.1 | 300 | 0.9 | 0 |
| CIFAR-100 | WRN28x10 | 0.1 | 200 | 0 | 0 |

## D.4 Details for Section 4.3

We use the code from [22][7] to calculate the eigenvalues/eigenvectors of the Hessian of a deep neural network using training data. We set `mode="lanczos"` to use the Lanczos algorithm. It uses the `scipy.sparse.linalg.eigsh` hook to the ARPACK Lanczos algorithm to find the top $k$ eigenvalues/eigenvectors using batches of data. We set `full_dataset=True` to use all data to calculate the eigenvalues.

Our goal is to find the top 10 positive eigenvalues and their associated eigenvectors. We use the default argument `which="LM"` in the Lanczos algorithm, which returns the top $k$ (assigned by the argument `num_eigenthings=k`) eigenvalues with the largest magnitude which may contain negative ones. In our experiment, $k = 30$ is large enough to contain the top 10 positive eigenvalues. Although the Lanczos algorithm supports method `"LA"` to directly return top $k$ positive eigenvalues, from our experience, the results are always significantly less than the top 10 positive eigenvalues chosen by the above procedure. We also replace the default `max_steps=20` to 1000 since in few cases the algorithm does not converge in 20 steps. We use the default tolerance `tol=1e-6`.

The DNNs used here are the same with those used in Section 4.2 with the same initializations and the same minibatches.

## D.5 Wall time and GPU memory consumption

In this section, we compare the wall time and the memory consumption between the gRDA and the SGD. All results in this section are done using the same server containing two Nvidia Tesla V100 (16GB memory) GPUs. We use two cards in training ResNet50-ImageNet, one card in training VGG16-CIFAR10 and WRN28x10-CIFAR100. Experiments are done serially. The training details are the same as described in Section D.1 and D.2. For the hyperparameters of the gRDA, we take $c = 0.001, \mu = 0.6$ for ResNet50-ImageNet, $c = 0.001, \mu = 0.4$ for VGG16-CIFAR10, and $c = 0.001, \mu = 0.501$ for WRN28x10-CIFAR100. The choice of $c$ and $\mu$ in gRDA does not affect the time usage and memory footprint.

For the wall time, in the case of ResNet50-ImageNet, we record the calculation time for the first 200 iterations per 10 iteration. We omit the first iteration since it is much larger than the others due to model initiation on GPU. We calculate the average and the standard deviation using the remaining sample of size 19. In the cases of VGG16-CIFAR10 and WRN28x10-CIFAR-100, we record the calculation time for the first 20 epochs (390 iterations per epoch) and omit the very first epoch. We calculate the mean and the standard deviation of the 19 trials.

For the memory consumption, we focus on the peak GPU memory usage, i.e. the maximum memory usage during training, since it determines whether the task is trainable on the given platform. In the case of ResNet50-ImageNet, we record the usage of the first 200 iterations among 5 different training tries. We show the memory usage for two GPU cards separately because the design of PyTorch leads to a higher memory usage in the card0. In the cases of VGG16-CIFAR10 and WRN28x10-CIFAR-100, we record the peak GPU memory usage throughout the first 20 epochs. We calculate the mean and the standard deviation of the 5 tries.

From Table 7, the gRDA generally requires a higher wall time than the SGD, because gRDA requires an additional step for the soft thresholding. For the memory consumption, one can observe that the difference between the gRDA and the SGD depends on the tasks and architectures, although it is

Figure 8: Learning trajectories of (`SGD`) and (`gRDA`) for VGG16 on CIFAR-10. See Section D.2 for the selection of hyperparameters about training.

generally small. In particular for the case of ResNet50-ImageNet, the difference in means of the SGD and the gRDA is not significant since it is less than their respective standard deviations. In fact, we find that the GPU memory consumption is unstable in these 5 tries, and sometimes the gRDA uses slightly less GPU memory than the SGD. The reason of the difference could be due to the underlying design of PyTorch, which may be interesting for future research.

Figure 9: Learning trajectories of (SGD) and (gRDA) for WRN28x10 on CIFAR-10. See Section D.2 for the selection of hyperparameters about training.

Figure 10: Learning trajectories of (SGD) and (gRDA) for WRN28x10 on CIFAR-100. See Section D.2 for the selection of hyperparameters about training.

Table 4: Details of the learning trajectories in Figure 8 at some selected epoch, which compare (SGD) and (gRDA) for VGG16 on CIFAR-10. The means and the standard deviations (in the parenthesis) are taken on 10 independent trials initialized with independent random initializers.

| Epoch | 1 | 50 | 100 | 200 | 300 | 600 |
|---|---|---|---|---|---|---|
| | | | Training Loss | | | |
| SGD | 2.1797(0.0859) | 0.1453(0.0091) | 0.0404(0.0081) | 0.0129(0.0017) | 0.007(0.0019) | 0.0(0.0) |
| gRDA(0.4) | 2.191(0.0897) | 0.1537(0.0097) | 0.0362(0.003) | 0.0124(0.0047) | 0.005(0.0014) | 0.0(0.0) |
| gRDA(0.51) | 2.2044(0.0939) | 0.149(0.0185) | 0.0351(0.0036) | 0.011(0.0031) | 0.0057(0.0018) | 0.0(0.0) |
| gRDA(0.6) | 2.1735(0.0817) | 0.1557(0.012) | 0.0392(0.0054) | 0.0169(0.0051) | 0.0096(0.0023) | 0.0001(0.0001) |
| gRDA(0.7) | 2.2394(0.0473) | 0.2262(0.0386) | 0.0644(0.0072) | 0.0304(0.0094) | 0.0223(0.0039) | 0.0002(0.0001) |
| | | | Training Accuracy (%) | | | |
| SGD | 16.6538(3.6064) | 95.1673(0.2846) | 98.666(0.2707) | 99.5902(0.0466) | 99.7911(0.0554) | 99.9993(0.0013) |
| gRDA(0.4) | 15.8911(3.8974) | 94.8553(0.3168) | 98.7909(0.1281) | 99.6102(0.1431) | 99.8367(0.0429) | 99.9993(0.0009) |
| gRDA(0.51) | 15.7698(3.9934) | 95.0562(0.6049) | 98.8393(0.1085) | 99.6491(0.1029) | 99.8244(0.0484) | 99.9996(0.0008) |
| gRDA(0.6) | 16.8571(3.6324) | 94.7998(0.415) | 98.7262(0.1691) | 99.4624(0.1419) | 99.6913(0.0658) | 99.9991(0.001) |
| gRDA(0.7) | 14.7902(3.0521) | 92.6507(1.1549) | 97.8649(0.2249) | 99.0433(0.2869) | 99.3093(0.1221) | 99.9973(0.0019) |
| | | | Testing Loss | | | |
| SGD | 2.1212(0.0866) | 0.4768(0.017) | 0.5667(0.0142) | 0.6561(0.0338) | 0.6843(0.0392) | 1.0713(0.0207) |
| gRDA(0.4) | 2.1185(0.0575) | 0.4966(0.0234) | 0.588(0.0314) | 0.6729(0.0463) | 0.7159(0.0267) | 0.9868(0.0372) |
| gRDA(0.51) | 2.1459(0.0956) | 0.5002(0.0367) | 0.5748(0.0217) | 0.6485(0.0294) | 0.6745(0.0288) | 0.793(0.0268) |
| gRDA(0.6) | 2.0925(0.0481) | 0.4856(0.0284) | 0.5709(0.0209) | 0.6005(0.0364) | 0.6388(0.043) | 0.7618(0.0262) |
| gRDA(0.7) | 2.1854(0.0459) | 0.5055(0.0493) | 0.567(0.0233) | 0.6104(0.0221) | 0.6412(0.0305) | 0.8248(0.0215) |
| | | | Testing Accuracy (%) | | | |
| SGD | 19.1611(3.6631) | 87.3144(0.3584) | 88.73(0.2176) | 89.4422(0.2907) | 89.8178(0.1843) | 90.8178(0.1779) |
| gRDA(0.4) | 19.01(2.6052) | 86.9711(0.4733) | 88.6333(0.3905) | 89.5444(0.2929) | 90.0522(0.2752) | 90.87(0.2082) |
| gRDA(0.51) | 18.4556(4.2166) | 87.0656(0.789) | 88.8511(0.3015) | 89.54(0.3037) | 90.0478(0.2855) | 91.01(0.1464) |
| gRDA(0.6) | 20.1911(2.4566) | 87.1733(0.6538) | 88.6722(0.4554) | 89.3278(0.4227) | 89.6856(0.4379) | 90.8244(0.2414) |
| gRDA(0.7) | 17.3589(2.4806) | 85.69(0.9893) | 87.7489(0.4771) | 89.1044(0.3582) | 89.2267(0.4733) | 90.6433(0.224) |
| | | | Sparsity | | | |
| SGD | 0.0(0.0) | 0.0(0.0) | 0.0(0.0) | 0.0(0.0) | 0.0(0.0) | 0.0(0.0) |
| gRDA(0.4) | 2.4922(0.0042) | 11.2733(0.0245) | 14.4122(0.0364) | 18.5011(0.0519) | 21.4367(0.0646) | 24.4433(0.0766) |
| gRDA(0.51) | 3.73(0.0) | 25.4678(0.0464) | 34.5056(0.0677) | 46.2378(0.0961) | 54.2222(0.1095) | 61.4322(0.1143) |
| gRDA(0.6) | 5.19(0.0) | 47.5333(0.0994) | 63.9333(0.169) | 79.6367(0.2028) | 86.3644(0.1928) | 90.49(0.1593) |
| gRDA(0.7) | 7.48(0.0) | 80.3589(0.225) | 92.0922(0.2057) | 95.5589(0.1481) | 96.39(0.109) | 97.2167(0.0767) |

Table 5: Details of the learning trajectories in Figure 9 at some selected epoch, which compare (SGD) and (gRDA) for WRN28x10 on CIFAR-10. The means and the standard deviations (in the parenthesis) are taken on 10 independent trials initialized with independent random initializers.

| Epoch | 1 | 25 | 50 | 75 | 100 | 200 |
|---|---|---|---|---|---|---|
| | | | Training Loss | | | |
| SGD | 0.9002(0.0056) | 0.0778(0.002) | 0.0177(0.0023) | 0.0078(0.0016) | 0.0047(0.001) | 0.0001(0.0) |
| gRDA(0.501) | 0.9087(0.005) | 0.0982(0.0014) | 0.0298(0.0017) | 0.0161(0.003) | 0.0124(0.003) | 0.0001(0.0) |
| gRDA(0.55) | 0.9112(0.0034) | 0.1101(0.0018) | 0.0359(0.0026) | 0.0202(0.0026) | 0.0143(0.0027) | 0.0001(0.0) |
| gRDA(0.6) | 0.9155(0.0027) | 0.1194(0.0015) | 0.0378(0.0022) | 0.0233(0.0017) | 0.0178(0.0052) | 0.0001(0.0) |
| gRDA(0.65) | 0.9162(0.0053) | 0.1251(0.002) | 0.0442(0.0024) | 0.0275(0.0026) | 0.0209(0.0027) | 0.0002(0.0) |
| | | | Training Accuracy (%) | | | |
| SGD | 67.9678(0.2445) | 97.2832(0.0693) | 99.398(0.096) | 99.7476(0.0537) | 99.8456(0.0387) | 99.9996(0.0008) |
| gRDA(0.501) | 67.7258(0.2829) | 96.5754(0.0475) | 99.0054(0.0772) | 99.4766(0.1011) | 99.587(0.1076) | 99.9998(0.0006) |
| gRDA(0.55) | 67.545(0.1816) | 96.1636(0.0878) | 98.7718(0.1102) | 99.3382(0.101) | 99.5198(0.0993) | 99.9986(0.0018) |
| gRDA(0.6) | 67.4044(0.1635) | 95.8476(0.0772) | 98.702(0.0927) | 99.2146(0.0636) | 99.3944(0.1851) | 99.9984(0.0012) |
| gRDA(0.65) | 67.42(0.3047) | 95.6366(0.0744) | 98.4866(0.0999) | 99.0694(0.0985) | 99.299(0.1049) | 99.9982(0.0017) |
| | | | Testing Loss | | | |
| SGD | 1.2606(0.1645) | 0.4845(0.0837) | 0.3849(0.0278) | 0.4238(0.0942) | 0.4151(0.0275) | 0.3624(0.0074) |
| gRDA(0.501) | 1.2986(0.2294) | 0.4099(0.0584) | 0.4008(0.0631) | 0.4547(0.0998) | 0.3918(0.0521) | 0.3241(0.0099) |
| gRDA(0.55) | 1.2489(0.1411) | 0.5151(0.193) | 0.3854(0.048) | 0.4738(0.1341) | 0.3899(0.0455) | 0.3266(0.0069) |
| gRDA(0.6) | 1.345(0.1042) | 0.4063(0.0352) | 0.3941(0.0632) | 0.4135(0.095) | 0.4414(0.1226) | 0.3245(0.0075) |
| gRDA(0.65) | 1.3211(0.216) | 0.4487(0.0556) | 0.346(0.0343) | 0.3805(0.0521) | 0.3873(0.0603) | 0.3182(0.0048) |
| | | | Testing Accuracy (%) | | | |
| SGD | 61.478(2.674) | 88.06(1.4363) | 91.978(0.3655) | 92.363(1.0626) | 92.968(0.3802) | 94.173(0.1127) |
| gRDA(0.501) | 61.021(4.0857) | 88.858(1.0066) | 91.161(0.9286) | 91.109(1.4645) | 92.517(0.7402) | 94.459(0.12) |
| gRDA(0.55) | 61.565(2.8849) | 86.546(2.9551) | 91.219(0.7465) | 90.717(1.6355) | 92.4(0.5953) | 94.433(0.0805) |
| gRDA(0.6) | 59.559(1.8942) | 88.481(0.9584) | 90.922(1.1019) | 91.486(1.2848) | 91.554(1.6144) | 94.497(0.166) |
| gRDA(0.65) | 59.934(3.7627) | 87.733(0.7991) | 91.637(0.5235) | 91.683(0.9814) | 91.996(0.8251) | 94.531(0.1119) |
| | | | Sparsity (%) | | | |
| SGD | 0.0(0.0) | 0.0(0.0) | 0.0(0.0) | 0.0(0.0) | 0.0(0.0) | 0.0(0.0) |
| gRDA(0.501) | 19.1382(0.003) | 64.4465(0.0217) | 77.8759(0.0513) | 82.8773(0.0502) | 85.5712(0.0425) | 90.1746(0.0301) |
| gRDA(0.55) | 23.6897(0.0022) | 79.9356(0.0209) | 87.2516(0.0189) | 90.0057(0.0223) | 91.5633(0.0357) | 94.7444(0.0367) |
| gRDA(0.6) | 29.4495(0.0013) | 88.8105(0.0315) | 92.5813(0.019) | 94.155(0.0219) | 95.0531(0.0178) | 97.0747(0.0179) |
| gRDA(0.65) | 36.6118(0.0024) | 93.5285(0.0143) | 95.6032(0.0127) | 96.5137(0.0111) | 97.0338(0.0138) | 98.2877(0.0125) |

Table 6: Details of the learning trajectories in Figure 10 at some selected epoch, which compare (`SGD`) and (`gRDA`) for WRN28x10 on CIFAR-100. The means and the standard deviations (in the parenthesis) are taken on 10 independent trials initialized with independent random initializers.

| Epoch | 1 | 25 | 50 | 75 | 100 | 200 |
|---|---|---|---|---|---|---|
| | | | Training Loss | | | |
| SGD | 2.9477(0.0037) | 0.3347(0.0038) | 0.0348(0.0044) | 0.0076(0.0023) | 0.0023(0.0028) | 0.0004(0.0) |
| gRDA(0.501) | 2.9632(0.0052) | 0.3921(0.0031) | 0.0652(0.0047) | 0.033(0.0073) | 0.0233(0.0063) | 0.0006(0.0) |
| gRDA(0.55) | 2.9752(0.0052) | 0.4167(0.0025) | 0.0758(0.005) | 0.0427(0.0055) | 0.0317(0.0077) | 0.0007(0.0) |
| gRDA(0.6) | 2.9798(0.0038) | 0.4407(0.0042) | 0.0966(0.0041) | 0.0542(0.0039) | 0.0394(0.0075) | 0.0008(0.0) |
| gRDA(0.65) | 2.9885(0.0062) | 0.4633(0.0035) | 0.1188(0.0046) | 0.0692(0.0053) | 0.0495(0.0038) | 0.0011(0.0) |
| | | | Training Accuracy (%) | | | |
| SGD | 25.7816(0.1029) | 89.2406(0.1343) | 99.071(0.1465) | 99.825(0.0637) | 99.9418(0.0723) | 99.9822(0.0038) |
| gRDA(0.501) | 25.4118(0.0721) | 87.4652(0.141) | 98.069(0.1749) | 99.0446(0.256) | 99.327(0.2105) | 99.9792(0.0022) |
| gRDA(0.55) | 25.188(0.1628) | 86.6598(0.0732) | 97.7116(0.1959) | 98.7498(0.2099) | 99.0674(0.2492) | 99.9794(0.0035) |
| gRDA(0.6) | 25.092(0.0661) | 86.0122(0.1322) | 97.0026(0.1456) | 98.3896(0.1561) | 98.8524(0.253) | 99.977(0.0029) |
| gRDA(0.65) | 24.9426(0.128) | 85.3238(0.1064) | 96.2586(0.1602) | 97.9104(0.1846) | 98.533(0.1325) | 99.9758(0.0014) |
| | | | Testing Loss | | | |
| SGD | 3.6524(0.1035) | 1.423(0.1287) | 1.5834(0.0411) | 1.5164(0.0427) | 1.5109(0.0621) | 1.4977(0.0164) |
| gRDA(0.501) | 3.6615(0.1535) | 1.3842(0.0697) | 1.7227(0.1908) | 1.646(0.2176) | 1.6685(0.1295) | 1.3766(0.016) |
| gRDA(0.55) | 3.8418(0.1471) | 1.3317(0.0558) | 1.7881(0.1633) | 1.6067(0.1051) | 1.6622(0.1452) | 1.3537(0.0204) |
| gRDA(0.6) | 3.8368(0.1623) | 1.332(0.0551) | 1.6861(0.1355) | 1.6166(0.0618) | 1.6807(0.1061) | 1.3789(0.0181) |
| gRDA(0.65) | 3.7701(0.1759) | 1.3689(0.1135) | 1.6668(0.1081) | 1.5665(0.1276) | 1.6499(0.1364) | 1.3722(0.0156) |
| | | | Testing Accuracy (%) | | | |
| SGD | 19.408(1.1196) | 66.605(1.8987) | 72.018(0.6428) | 74.775(0.4029) | 75.807(0.638) | 76.529(0.169) |
| gRDA(0.501) | 18.798(1.2235) | 66.621(1.0782) | 69.549(1.7699) | 71.9(1.7964) | 72.47(1.2624) | 76.916(0.1894) |
| gRDA(0.55) | 17.644(0.9865) | 67.039(0.8587) | 68.437(1.6978) | 72.095(1.3079) | 71.792(1.8297) | 76.996(0.1713) |
| gRDA(0.6) | 17.761(0.9144) | 67.107(1.0409) | 69.13(1.6105) | 71.334(0.6645) | 71.328(0.9034) | 76.999(0.3635) |
| gRDA(0.65) | 17.653(1.0898) | 66.13(2.0093) | 68.698(1.1661) | 71.416(1.5185) | 71.3(1.5667) | 76.853(0.1996) |
| | | | Sparsity (%) | | | |
| SGD | 0.0(0.0) | 0.0(0.0) | 0.0(0.0) | 0.0(0.0) | 0.0(0.0) | 0.0(0.0) |
| gRDA(0.501) | 19.1087(0.0031) | 58.3232(0.0284) | 67.0737(0.0783) | 72.4028(0.0876) | 75.8975(0.1454) | 82.6918(0.0986) |
| gRDA(0.55) | 23.6534(0.0026) | 71.3479(0.0412) | 77.8894(0.0343) | 81.8828(0.0499) | 84.2722(0.0676) | 90.2043(0.0508) |
| gRDA(0.6) | 29.4058(0.0024) | 81.4373(0.0293) | 85.6999(0.0211) | 88.3771(0.0221) | 89.9111(0.0318) | 94.2045(0.0649) |
| gRDA(0.65) | 36.5536(0.002) | 88.195(0.0166) | 90.845(0.0191) | 92.5057(0.018) | 93.4805(0.0184) | 96.3042(0.0219) |

Table 7: Comparison of SGD and gRDA on time and GPU memory consumption. The values in the upper penal of the table are the average time consumption of 19 records excluding the initial iterations. The values in the lower penal are the average peak GPU memory consumption of 5 different tries. The numbers in parenthesis are the standard deviation.

| | Time per iteration (s) | | |
|---|---|---|---|
| | ResNet50-ImageNet | VGG16-CIFAR10 | WRN28x10-CIFAR100 |
| SGD | 0.3964 (0.0183) | 0.0214 (0.0002) | 0.2271 (0.0008) |
| gRDA | 0.4582 (0.0166) | 0.0303 (0.0004) | 0.2510 (0.0011) |
| | GPU Memory (MiB) | | |
| | ResNet50-ImageNet (card0,1) | VGG16-CIFAR10 (card0) | WRN28x10-CIFAR100 (card0) |
| SGD | 14221 (376), 14106 (380) | 1756 (48.6) | 10301 (0) |
| gRDA | 14159 (167), 13947 (208) | 1809 (10.2) | 10589 (0) |

(a) VGG16/CIFAR-10/Train loss/$\mu = 0.4$

(b) VGG16/CIFAR-10/Test error/$\mu = 0.4$

(c) VGG16/CIFAR-10/Train loss/$\mu = 0.51$

(d) VGG16/CIFAR-10/Test error/$\mu = 0.51$

(e) VGG16/CIFAR-10/Train loss/$\mu = 0.7$

(f) VGG16/CIFAR-10/Test error/$\mu = 0.7$

(g) WRN28x10/CIFAR-100/Train loss/$\mu = 0.55$

(h) WRN28x10/CIFAR-100/Test error/$\mu = 0.55$

(i) WRN28x10/CIFAR-100/Train loss/$\mu = 0.6$

(j) WRN28x10/CIFAR-100/Test error/$\mu = 0.6$

(k) WRN28x10/CIFAR-100/Train loss/$\mu = 0.65$

(l) WRN28x10/CIFAR-100/Test error/$\mu = 0.65$

Figure 11: The contour of training loss and testing error around the minimal loss/error curves. (Figure 5) The right end point is the SGD, and the left end point is the gRDA.

[Supplementary Material 2 · gRDA_SGD_cifar10_vgg16_10seeds_acc_loss.pdf]

Legend:
- SGD (no other parameters)
- gRDA: c=0.0005, mu=0.4
- gRDA: c=0.0005, mu=0.51
- gRDA: c=0.0005, mu=0.6
- gRDA: c=0.0005, mu=0.7

Vertical axes (top to bottom): train loss, train accuracy, test loss, test accuracy, sparsity. Horizontal axis: epoch.

[Supplementary Material 3 · cifar100_wideresnet_small.pdf]

Compare SGD with gRDA
(Data: CIFAR100, Model: WideResNet28x10, lr=0.1)

Legend:
- SGD (no other parameters)
- gRDA: c=0.001, mu=0.501
- gRDA: c=0.001, mu=0.55
- gRDA: c=0.001, mu=0.6
- gRDA: c=0.001, mu=0.65

[Supplementary Material 4 · imagenet_training-largefont.pdf]

Legend:
- gRDA: c=0.01, mu=0.4
- gRDA: c=0.005, mu=0.501
- gRDA: c=0.005, mu=0.51
- gRDA: c=0.005, mu=0.55
- SGD (no other parameters)