[Reviews · NeurIPS 2020]

Review 1

Summary and Contributions: gRDA [8] is a generalized formulation of regularized dual average method (RDA), which optimizes the dense neural networks to have sparse parameters. The authors propose a directional pruning which compensates the training loss due to weight pruning by manipulating unpruned weights slightly, and then show that their method is effectively equivalent to gRDA under loose assumptions. Besides, they evaluated the performance of [8] under various architecture and datasets and show that gRDA is comparable to other pruning methods.

Strengths: They provide an intuitive explanation on why gRDA is effective in the perspective of pruning using the proposed directional pruning. In competitive setting, the authors empirically justify that gRDA is effective in pruning. On ImageNet, when # of parameters is smaller than 4 Million (quite extreme condition), they reported 73-77% test accuracy and this is comparable to the state-of-the-art dense networks such as EfficientNet or RegNet.

Weaknesses: - Based on my understanding, directional pruning does not resolve the issues in the previous work. Authors criticized other magnitude based pruning because existing MP methods require expert knowledge or trial-and-error based learning. However, such an issue has been resolved recently. For example, [17] and related (lottery ticket hypothesis related) family fix the pruning rate as 20% consistently. I rather think that the proposed method is sensitive to c or mu value, thus it requires manual tuning to reach the desired level of sparsity. - Despite the authors used gDRA for pruning and compared gDRA with others, the effectiveness of gDRA is neither the novelty or contribution of this work. Thus, regardless of how good gRDA is, it should be separated from the value of this study. However, the authors relate gRDA and the proposed method by emphasizing similarity between gRDA and directional pruning and then show that gRDA has a powerful pruning performance. This misleads the readership as if directional pruning has a powerful pruning performance, although it is not. Especially, I think the expressions in Line 97 and 234, ‘our algorithm~’ can really mislead the readership.

Correctness: Probably right.

Clarity: Yes, except that the part we mentioned on the weakness. Also, I found that some reference paper is incorrect-in Line 217, [46] does not conduct CIFAR-10 experiment although the authors said they did.

Relation to Prior Work: Yes

Reproducibility: Yes

Additional Feedback:


Review 2

Summary and Contributions: This paper proposes the concept of directional pruning -- pruning along directions of little to no increase is the loss surface -- as well as a method to implement directional pruning using gRDA.

Strengths: - Important problem; the theoretical and practical aspects of pruning has received increased attention. The existing landscape of techniques shows muddled results so more clarity is necessary.

Weaknesses: The scope of the contributions themselves are unclear, particularly with respect to previous work. I elaborate on this in more detail below.

Correctness: Claims are accurate as far as I can tell. The empirical methodology could use improvements in the execution of the paper. For example, the paper includes at multiple points that "retraining is not required" however the presentation of the evaluation suggests that this is -- in essence -- a regularized learning technique with the full approach beginning with a random initialization. This does not fit retraining in the sense of techniques like OBS or magnitude pruning which are applied to a completely trained network and therefore may require retraining after pruning. I suggest removing these statements from the paper if my analysis is accurate. Another issue is this comment about the efficacy of the technique with standard hyperparameters: > L597: We found that the gRDA does not yield a competitive result under this learning rate schedule. It is unfortunate -- and somewhat misleading -- to only have this in the Appendix because it's important to understand how the techniques work with standard hyperparameters.

Clarity: Overall, the imprecision of the claims with the lack of a direct comparison of OBD's and directional pruning's analysis make this paper's presentation less clear than optimal. Specifically, I am very unclear on the shape of $\mathcal{P}_0$. The $s_j$ in Eq. 3 denote the distance between $w$ and $\mathcal{P}_0$. However, $\mathcal{P}_0$ has a very ambiguous definition as the subspace of directions with nearly 0 eigenvalues. The "nearly 0" suggests some manner of thresholding at play and the developments in Sections 1.1, 2, and 3 do not seem to make this explicit. Theorem 3.1 asserts convergence in the limit of $\overline{s}_j$ to appropriate $s_j$ but I'm still left wondering exactly what $\mathcal{P}_0$ is in any explicit fashion, such as in the form of bounds on the eigenvalues of the directions that are included. A3 separates out $\overline{H}$ as this subspace of exactly zero eigenvalues but this seems like an optimistic assumption, with $\mathcal{P}_0$ generally including directions with nonzero eigenvalues. Please clarify in the response if possible.

Relation to Prior Work: I would deeply appreciate a restated set of contributions in the author response. Specifically, the claimed contributions of a method appear to be boxed in by [8] and the overall directional pruning concept (and analysis in 1.1) is boxed in by OBS (and other methods). To clarify, (1.2) in 8 is (to the best of my reading) that same as (5) in this paper (as noted by Line 117). Therefore the algorithm is pre-existing I believe in whole, which is not made 100% clear. Specifically, multiple points in the abstract and 1.1 refer to "our algorithm." A second read of 1.2 makes it more clear that the paper does indeed acknowledge that this is an existing algorithm. However, I highly suggest clarifying this earlier in the paper and reinforcing this language later on (e.g., L109, change "we adopt" to "we use"). The relationship to OBS is also a bit hazy because 1.1 relies on the definition of $s_j$ which has an imprecise definition in (2) via $\mathcal{P}_0$ whose value is not precise given (i.e., 'Perturbation in these directions causes little change ... denote the subspace generated by these directions a"). OBS provides an optimal solution (up to a second-order approximation of the loss) subject to the assumption that a network is at a minimum (and therefore first-order components are zero). Is it possible to demonstrate a precise relationship between the results here and those of OBS? Remark 1.3 notes that the optimization problems are different, but I'm unclear on to what extent this work differs.

Reproducibility: Yes

Additional Feedback: ### Comments My overall sense about this paper is that there is an interesting result here that would be significantly improved if the relationship to OBS were clarified, $\mathcal{P}_0$ were clarified, and the empirical results held up stronger. Particularly, on the last point, given that the method seems to not work particularly well with standard hyperparameters, I am less enthusiastic about directional pruning as a valuable pruning definition even though it seems natural. The results presented in the main body of the paper with non-standard hyperparameters and reduced accuracy for the initial network give me pause as well and, so perhaps the methodology of these experiments could be improved as well. Also, an alternative narrative that would make for a stronger result -- if true -- would be to map the OBS objective to solutions of this algorithm. In which case a reader needs not be concerned about if directional pruning itself is a valuable concept as OBS is already well established. Then, the proposed benefit of not needing a Hessian approximation (Remark 1.3) would then be the central value proposition. For the evaluation, the technique could then be compared to OBS frameworks with other Hessian approximations, ideally yielding improved results or reduced computation time. #### Suggestions Line 210: >Interestingly, the neighborhood of the midpoint on the Bézier curve often has a higher testing accuracy. I suggest citing the work on weight averaging: "Averaging Weights Leads to Wider Optima and Better Generalization. Pavel Izmailov, Dmitrii Podoprikhin, Timur Garipov, Dmitry Vetrov, Andrew Gordon Wilson. Uncertainty in Artificial Intelligence (UAI), 2018 #### After Response Thank you for the clarifications. I've upped my score based on the author's feedback and the hope that directional pruning works out. Particularly, I have some -- albeit limited -- hope for the directional pruning concept. Specifically, new pruning concepts and generally new sparse learning algorithms are important to investigate. The value proposition of a sparse training technique with no pretraining and theoretical guarantees is very very high. What holds me from a higher vote is that I would appreciate more evidence on the relation between this abstraction and existing concepts (e.g., OBS). Particularly, this space is cluttered by recent sparse training algorithms (e.g., l0 regularization and various approximations [1, 2]) so this is another. Though, unlike some of those others, it does enjoy theoretical guarantees. To the results themselves, I have a couple suggestions: 1) Compare against [19]’s “magnitude pruning v2” instead of the vanilla magnitude pruning. The reported results undereport where magnitude pruning seems to lie, particularly for sparsities whose accuracy match the original, dense networks. 2) The paper asserts that its results are "competitive." The claim I’d like to see in the paper is that "the technique shows promise, producing accuracy and sparsity trade-offs that are within range of contemporary techniques." Essentially, I do not think these are “competitive” in the sense that the sensitivity of the overall technique to the hyperparameters leads me to believe that the technique isn't particularly robust; that seems to be a shared concern. [1] Learning Sparse Neural Networks through L0 Regularization. Christos Louizos, Max Welling, Diederik P. Kingma. ICLR, 2018 [2] Soft Threshold Weight Reparameterization for Learnable Sparsity. Aditya Kusupati, Vivek Ramanujan, Raghav Somani, Mitchell Wortsman, Prateek Jain, Sham Kakade, Ali Farhadi. ICML, 2020.


Review 3

Summary and Contributions: The paper presents an algorithm for sparsifying neural network models. The basic idea is to find a sparse solution around minima by searching on a projected space (so is called “directional pruning”) that minimally changes the loss (where eigenvalues of Hessian are nearly zero). A sparse solution is found by solving an optimization objective iteratively with a l_1 proximal operator using a previously studied algorithm called gRDA. The proposed method finds a very sparse solution that performs well on image classification tasks (including relatively large-scale one; imagenet) and is demonstrated to be located in the similar loss surface of solution found by SGD. This is a great work, and the paper deserves acceptance.

Strengths: The proposed method is well-motivated; network sparsification is a promising approach to reduce computational requirements on deep learning, and doing so without increasing loss (so that it doesn’t require a retraining process) makes sense. The proposed method is principled; the idea of moving on the projection space where the loss shouldn’t increase based on the second order approximation makes sense, and further, doing so without explicitly relying on computation of Hessian is nice. The proposed method is novel; although the basic idea is not entirely new, this paper presents an approach that puts high contrast to many modern approaches for pruning neural networks that are often based on ad-hoc intuitions. The paper is written well; references to related works fit relevantly and are adequately placed (although can improve; see Relation to prior work); concise and accurate descriptions of mathematical/technical expressions. The experiments conducted are well suited to demonstrate the effectiveness of the proposed idea, and the results are convincing enough to support the claims being made throughout the paper.

Weaknesses: The preliminary material (i.e., gRDA) which the main method is based on can be polished to help the audience understand better the proposed method: the technical expressions become a bit loose in Section 2; e.g., the loss function l(w) is already defined earlier but another L(h;y) appears; state explicitly the iterates of w in gRDA to explain the algorithm; making c=0 in gRDA doesn’t bring SGD precisely/directly as W^SGD_n doesn’t appear in gRDA. These are perhaps minor issues, but because of these glitches the section doesn’t link to the solution for directional pruning in Section 1.1. at once.

Correctness: I believe so. I have not checked thoroughly the proofs in appendix though.

Clarity: The paper is written very well.

Relation to Prior Work: Overall, the paper discusses previous works well in terms of how they are related and different. For the sake of improvements, however, the reviewer would like to point out a bit of inaccurate descriptions as in the current manuscript and suggest a few missing works as follows: 1. (related to the claim that the proposed method is automatic and doesn’t require expert knowledge) considering that c and mu are indeed hyperparameters that can’t be easily tuned without expert knowledge. Therefore, it is unfair to claim improvement over previous work while describing them with threshold requirement or “trial-and-error” (L33), 2. (on the retraining requirement in L35) there are approaches that do not require retraining; in fact, almost all optimization based approaches (e.g. LC algorithms in Carreira-Perpinan and Idelbayev 2018) or pruning-while-training approaches (so-called dynamic pruning) do not require retraining by nature; also recently proposed pruning-before-training approaches (e.g. SNIP in Lee et al. 2019) do not require retraining by nature. Importantly, notice also that all these paradigms do not require pre-training (again by design), whereas the proposed method requires one (i.e. w^SGD). A small bit of discussion over this matter seems necessary.

Reproducibility: Yes

Additional Feedback: At first I doubted the level of sparsity this method could possibly achieve without losing accuracy. Based on seemingly quite successful solutions found, it appears that neural networks are indeed highly overparameterized. Have authors experienced *any* difficulty when finding extremely sparse solutions (potentially higher than what’s presented)? What happens when the method is applied to relatively low or moderately parameterized models (VGG and WRN are very overparameterized models)? If not tested or can't be tested within time, can authors provide some expectations, and further, how to mitigate the potential issue? Why do the solutions found at relatively lower sparsity levels perform noticeably worse than ones by other approaches (in Figure 4)? Is it because the algorithm is tuned with hyperparameters that are optimally set for higher sparsity? Can authors provide some intuitive comparisons (not necessarily need to run them if time is limited) or comments to different pruning approaches besides pruning-after-convergence as proposed? *post author response phase* The paper is motivated well (\ie, decay weights, or prune, in a direction that avoids increase in training loss), and the authors executed the suggested idea in a novel way (\eg, although gRDA was an existing algorithm, the use of it so as to avoid the need of directly estimating Hessian or its inverse is considered a non-trivial contribution at least in the context of network pruning literature), and achieved results to support their idea (\eg, the solutions found by directional pruning demonstrate competitive sparsity and performance) in an impressive manner (\eg, 1. their results are proved to be asymptotically correct, 2. while costing only slightly higher than SGD). The paper is written well with case. I did/do have concerns though, which are the followings: 1 (on re-training and pre-training) the paper would be much stronger if the authors could include discussion on this along with experimental results, 2. (hyperparameters) I understand tuning of c and mu could be relatively minor issues considering other works introducing many hyperparameters, yet still, the paper is unfairly inflating this aspect (e.g. “automatic”), and 3. (on overparameterized networks) the suggested algorithm would be more convincing if it is demonstrated to be applicable for more difficult cases (less parameters).


Review 4

Summary and Contributions: In this paper, they propose to prune weights without affecting the training loss. The pruning directions are thus the flat directions on the loss surface. Following this lead, the authors propose gRDA to prune weights along these flat directions. The authors provide theoretical analysis on gRDA and show that gRDA performs directional pruning.

Strengths: Though pruning weights along flat directions is not entirely new, existing algorithms usually need Taylor expansion to do so, which requires the Hessian matrix. In their paper, they successfully connect gRDA algorithm with pruning along with flat directions and do not require any second-order information. The theoretical analysis on how to connect gRDA with directional pruning is the major contribution of this paper. Moreover, gRDA algorithm itself is very simple, which just applies soft-thresholding on the accumulation of gradients with two hyperparameters c and /mu. I did not thoroughly examine the proof since I am not familiar with the ODE part. Besides theoretical analysis, the experimental results on ImageNet seem promising and the visualizations of pruning directions are also interesting.

Weaknesses: Since I am not familiar with the ODE, I will mainly focus on the weakness of the algorithm and experimental results. In section 1, the analysis is based on SGD (seems without momentum and weight decay). In Algorithm.1 and 2, as well as the gRDA update rule, it seems that only the accumulation of gradients (without momentum and weight decay) is considered. This setting contrary to the widely used SGD with momentum and weight decay scheme. If the gRDA can not cope with momentum and weight decay, it may restrict the pruned model to achieve its best performance. For example, in Figure. 7, when pruning 80% of weights, the model can only achieve 73% Top-1 accuracy, which is much lower than AMC. Usually, when removing 80% of weights, the model can maintain its performance which is around 76% Top-1 accuracy for a regular trained ResNet-50 model. Also in the state of sparsity [19], non-uniform magnitude pruning can reach 75.16% Top-1 accuracy with 90% sparsity, which seems better than gRDA. I do not mean to criticize the performance of gRDA, but there is an upper bound (around 73% accuracy) for the performance of gRDA. This limitation makes gRDA not a good choice for middle to high sparsity, probably due to no momentum and/or weight decay. gRDA requires a longer training time compared to regular training, and it can not use the original training scheduler, which may also relate to the absence of momentum and/or weight decay. The hyperparameter tuning of gRDA is not trivial. g(n, \gamma) = c\gamma^0.5(n\gamma)^\mu contains 2 hyperparameters \mu and c, and it also involves learning rate \gamma which might be changing. Thus, the hyperparameter setting is more complex compared to regular Lasso which only requires one hyperparameter. Although large \mu implies larger sparsity, the relationship is not that clear as shown in Figure.4 and Table.1 in appendix C. After Rebuttal: ------------------------------------------------- I have read the authors' rebuttal and other reviews. I appreciate the new experiments of adding momentum and weight decay. The gRDAM algorithm indeed improves the orignal results to some extent. The rebuttal addressed most of my concerns. Thus I will increase my score from 6 to 7. Another interesting property of gRDA is that it can achieve a quite low sparsity after a short time when training started (Figure.3 right). As stated in the state of sparsity [19], 'Re-training then can be done fully sparse, taking advantage of sparse linear algebra to greatly accelerate time-to-solution'. If the sparse topology is stable in gRDA, then it can enjoy such benefits. I think the authors could also add some discussions on whether the topology is stable during training, which may have practical impacts.

Correctness: The emprical results are correct, the theortical results seems reasonable, but I did not fully examine the proof.

Clarity: Yes, overall the writthing quality is good. It would be better, if Algorithm.1 and/or 2 can be moved into the main context.

Relation to Prior Work: Yes, it provides sufficient discussions.

Reproducibility: Yes

Additional Feedback: What is the reason for not including the momentum and/or weight decay in the theoretical analysis? Can it be included in the algorithm (even it's not included in theoretical analysis)? Seems that not using momentum and weight decay largely limits the potential performance of gRDA, especially for middle sparsity rate, say 80%. As training proceeds, g(n, \gamma) will be increased. Dynamic g(n, \gamma) may result in a sparser or denser model during training as shown in Figure 3 right. As a result, is there any principal way to determine when to stop training? In Table.1, the results of the regular training schedule is much worse compared to gRDA, what might be the reason behind this phenomena? It's possible to apply gRDA on pretrained models? Pruning pretrained models is a very common scenario considered in many weight pruning works.

[Author Response · NeurIPS 2020]

1 We thank all reviewers for their comments, which are in *italic* below. The number(s) in each item is the reviewer(s)
2 being addressed to. **CR** is the shorthand for camera-ready. Citation number here aligns with the references of the paper.

3 1. *Hyperparameters $c, \mu$ still require tuning (**1,3,4**).* The expert knowledge in our paper refers to knowing a good
4 sparsity level specific to the task and network. Tuning gRDA can be easier than prescribing the level of sparsity
5 (which gRDA does not require) when dealing with a new dataset/network with few prior research to base on. We
6 develop a rule of thumb to select $c, \mu$ (not in the original gRDA paper [8]). Our empirical results and theory suggest
7 $\mu \in \{0.501, 0.51, 0.55\}$ generally performs well regardless of the task and network used. For a given $\mu$, we search for
8 the greatest $c$ (starting with e.g. $10^{-4}$) such that gRDA yields a comparable test acc. as SGD using $1-5$ epochs.

momentum=0.9, wd=1e-4, (test acc@ep90)
gRDAM: c=0.005, mu=0.65, (72.39%)
gRDAM: c=0.01, mu=0.501, (74.33%)
gRDAM: c=0.0001, mu=0.501, (76.27%)
SGD, (76.02%)

2. *gRDA does not perform well under the PyTorch learning rate (lr) schedule; adding momentum and weight decay (WD) (**2,3,4**).* The PyTorch lr schedule for the ImageNet is usually applied jointly with the SGD with Polyak's momentum. Without it, SGD only yields a test accuracy of 68.76% for ImageNet-ResNet50 under this lr schedule. The current gRDA has no momentum, so its performance in the upper panel of Table 1 seems a little disappointing despite the extremely high sparsity (will revise Sect. 4.1 to include this). The absence of momentum could also be a reason for the underperformance at the middle to high sparsity level in Figure 4 (R3,R4). R4 suggests to include momentum and WD in gRDA.

19 We try Polyak's momentum and WD with the original $g(n, \gamma)$, termed gRDAM, on ResNet50 with ImageNet (figure on
20 left). While the sparsity reaches 88% and the test acc. improves, the test acc. of gRDAM still generally jumps less than
21 SGD with momentum when the lr drops. However, gRDAM with $c = 10^{-4}$ yields a higher test acc. than SGD but with
22 only 14.25% sparsity. Indeed, there is still much room for improvement which calls for a new theory to find a proper
23 $g(n, \gamma)$. Other momentum, e.g. Nesterov and QHM (Ma and Yarat, 2019, ICLR) can also be considered.

24 3. *Contributions (**1,2**).* We provide the first systematic study on the effectiveness of gRDA for pruning modern DNNs
25 on large-scale tasks, while the original gRDA paper [8] has not focused on deep learning. Inspired by the good empirical
26 performance of gRDA, we theoretically study gRDA and discover that it asymptotically performs the directional pruning
27 (DP, see below for a comparison to OBS), which is empirically verified by the connectivity (Sect. 4.2) and subspace
28 restriction (Sect. 4.3). This justifies a unified view of gRDA and DP. See 1. for a response on the expert knowledge.

29 4. *The shape of $\mathcal{P}_0$ and a comparison with the "optimal brain surgeon"(OBS)(**2**).* We will revise Sect. 1.1 to define $\mathcal{P}_0$
30 as the eigenspace corresponding to the zero eigenvalues of the Hessian $H(\mathbf{w}(\infty)) = \nabla^2 \ell(\mathbf{w}(\infty))$ where $\mathbf{w}(t)$ is the
31 gradient flow and $\mathbf{w}(\infty) = \lim_{t \to \infty} \mathbf{w}(t)$ that achieves the minimum (under weak conditions) where flat directions
32 exist under overparameterization. This $\mathcal{P}_0$ is the eigenspace of zero eigenvalues of $\bar{H}$ as in (A3). Perturbation from
33 $\mathbf{w}^{SGD}$ along $\mathcal{P}_0$ causes little changes to loss $\ell$ if $\mathbf{w}^{SGD}$ reaches the same minimum valley as $\mathbf{w}(\infty)$, which holds under
34 a small learning rate. An analytic map between DP and OBS is interesting for future study, and we believe the two
35 are generally nonequivalent. Particularly, DP perturbs from $\mathbf{w}^{SGD}$ continuously in $\lambda$ like a restricted $\ell_1$ weight decay
36 on $\mathcal{P}_0$ (Remark 1.2), while OBS yields a discontinuous perturbation like a hard thresholding (see OBS, p.165 [29]).

using 30 epochs
gRDAM (mu=0.501)
GlobalMagWeight (Adam)
LayerMagWeight (Adam)
RandomPruning (Adam)

5. *Re-training and pre-training (**2,3,4**).* We agree with R2 and R3 and will revise the tone about re-training in the CR. For R3's question, directional pruning (DP) does not require pre-training with SGD, as gRDA achieves that in one shot training from scratch (shown in Eq. (8) in Thm 1). Although generally not recommended, gRDA can be implemented on the pre-trained models (R4). For an illustration, we re-train ResNet20 on CIFAR-10 using the gRDAM on a pre-trained model (ShrinkBench by Blalock et al., arXiv:2003.03033) and compare with their Fig. 11 using their codes and setting. The results (on the left) show that gRDAM outperforms several

46 magnitude pruning based methods under high compression level. To further improve, we should modify $g(n, \gamma)$; see 2.

47 6. *Previous work of gRDA and incorrect/missing references (**1,2,3**).* We will be more careful on referring gRDA; e.g, in
48 Line 97, 234 (R1) and 109 (R2). Sect. 1 will be revised to discuss the gRDA (R2,R3). The [46] in Line 221 should have
49 been Papyan (2018, arXiv:1811.07062) (R1); Izmailov et al. (2018, UAI) will be cited in Line 210 in the CR (R2).

50 7. *Notational confusions and $c = 0$ in gRDA (**3**).* Note $\ell(w) = N^{-1} \sum_{i=1}^{N} \mathcal{L}(h(X_i; \mathbf{w}), Y_i)$, where $(X_i, Y_i)$'s are the
51 training data, so $\ell(\mathbf{w})$ and $\mathcal{L}$ are different. As $c = 0$, gRDA is unpenalized, so it reduces to SGD (Eq. (8) in Thm 1).

52 8. *Principle to stop training (**4**).* We stop training when the test acc stabilizes. While the sparsity usually also stabilizes
53 at a high level with the test acc like in CIFAR-10/100, sometimes it can slowly decrease like in ImageNet-RN50 in
54 Figure 3. However, given that the level of sparsity in Figure 3 is still high, the concern should be minor.

55 9. *Additional comments of R3 (**3**).* The main challenge in implementation is tuning $c, \mu$ as addressed in point 1. For a
56 small network like ResNet20 on CIFAR-10, gRDA with $(c, \mu) = (0.01, 0.7)$ achieves test acc/sparsity 90.19/90.46%,
57 while SGD (w/o momentum & WD) achieves a test acc. of 89.12%, so it is still overparameterized. We leave a decent
58 comparison of pruning methods to future study due to its independent interest (Blalock et al., 2020, arXiv:2003.03033).

[Meta-Review · NeurIPS 2020]

Thank you for your submission. There were many internal discussion about the paper. R3 championed the paper and appreciated the fact the method has theoretical footing. R1 & R2 raise critical issues with the empirical evaluation. R1 correctly highlighted that experiments do not include important baselines. Additionally, the evaluation was done on a nonstandard learning rate schedules, and results on standard learning rate schedule are not fully convincing (feedback didn’t resolve this issue). R1 and R2 were also not convinced about hyperparameter selection. However, R2, R3, R4 found theoretical results important. Reviewer R2 raised the score based on the value proposition of the theoretical results. Based on the value of theoretical results I am happy to accept the work. However, this is conditional on addressing reviewers comments, and please pay special attention to R2 comments: (1) comparing to magnitude prunning v2, (2) changing wording around “competitive results” to something along the lines of “the technique shows promise, producing accuracy and sparsity trade-offs that are within range of contemporary techniques”. Please also include a detailed discussion of how easy it is to tune the hyperparameters.